# TRAINING-FREE DATASET PRUNING FOR INSTANCE SEGMENTATION

**Yalun Dai**[1,2,3]**, Lingao Xiao**[1,2,4]**, Ivor W. Tsang**[1,2,3]**, Yang He**[1,2,4]*

[1]CFAR, Agency for Science, Technology and Research, Singapore
[2]IHPC, Agency for Science, Technology and Research, Singapore
[3]Nanyang Technological University, [4]National University of Singapore
`daiy0018@e.ntu.edu.sg`, `xiao_lingao@u.nus.edu`
`{Ivor_Tsang, He_Yang}@cfar.a-star.edu.sg`

## ABSTRACT

Existing dataset pruning techniques primarily focus on classification tasks, limiting their applicability to more complex and practical tasks like instance segmentation. Instance segmentation presents three key challenges: pixel-level annotations, instance area variations, and class imbalances, which significantly complicate dataset pruning efforts. Directly adapting existing classification-based pruning methods proves ineffective due to their reliance on time-consuming model training process. To address this, we propose a novel **T**raining-**F**ree **D**ataset **P**runing (**TFDP**) method for instance segmentation. Specifically, we leverage shape and class information from image annotations to design a Shape Complexity Score (SCS), refining it into a Scale-Invariant (SI-SCS) and Class-Balanced (CB-SCS) versions to address instance area variations and class imbalances, all without requiring model training. We achieve state-of-the-art results on VOC 2012, Cityscapes, and COCO datasets, generalizing well across CNN and Transformer architectures. Remarkably, our approach accelerates the pruning process by an average of **1349**× on COCO compared to the adapted baselines. Source code is available at: https://github.com/he-y/dataset-pruning-for-instance-segmentation.

## 1 INTRODUCTION

Current dataset pruning methods (Coleman et al., 2019; Toneva et al., 2019; Tan et al., 2023) focus on image classification tasks, while neglecting more complex and practical tasks such as instance segmentation. Classification is relatively simple, typically dealing with images containing one primary object. In contrast, instance segmentation faces greater challenges, handling real-world images with multiple objects of varying classes, areas, and positions within a single image.

In this paper, we address dataset pruning for instance segmentation and identify three unique challenges. 1) *Pixel-level annotations.* Unlike classification tasks where each image has a single one-hot category label, instance segmentation tasks require labeling each pixel of an image, often with multiple different category labels present in a single image (Lin et al., 2014). 2) *Variable instance areas.* While images in classification tasks generally deal with images of consistent resolution, instance segmentation involves objects of varying areas within the same image. Fig. 2a and Appendix B demonstrates this diversity in multiple datasets, aligning with the observations in

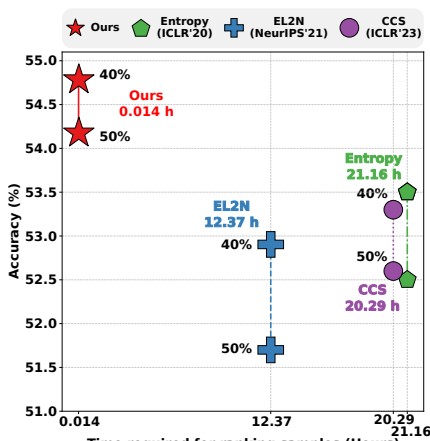

Figure 1: Comparison of $AP_{50}$ and runtime for sample ranking on COCO dataset: Our method vs. Entropy, EL2N, and CCS at 40% and 50% pruning rates. Our approach demonstrates superior efficiency and accuracy.

---

*Corresponding Author.
This work was completed during Yalun Dai's internship at CFAR, A*STAR.

(Lin et al., 2014). 3) **Class imbalance.** In contrast to classification tasks where the image count of each category is typically uniform, instance segmentation (as shown in Fig. 2b) inherently contains imbalanced object counts across classes. This imbalance stems from the natural image collection process, which often captures multiple objects of varying frequencies in real-world scenes (Oksuz et al., 2020). Addressing these challenges is crucial for advancing instance segmentation dataset pruning.

Additionally, existing dataset pruning methods face a fundamental contradiction: they often require a time-consuming model training process to identify important samples despite aiming to reduce overall training time. This paradox stems from several fundamental issues inherent in existing methods: 1) These methods often necessitate training on the entire dataset to calculate relevance scores (Paul et al., 2021; Coleman et al., 2019; Pleiss et al., 2020; Toneva et al., 2019), negating the intended time-saving benefits. 2) Samples selected based on a single model's output often show limited generalization to models with different architectures (Yang et al., 2023), further compromising efficiency gains. 3) In real-world scenarios, model training may be infeasible due to insufficient resources (Khouas et al., 2024), rendering existing methods inapplicable to these scenarios.

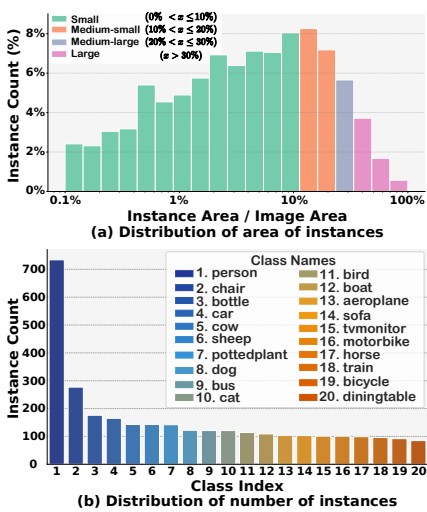

Figure 2: Visualization of VOC 2012 dataset to show variable instance area (a) and class imbalance (b).

To address the problems mentioned above, we propose a novel **T**raining-**F**ree **D**ataset **P**runing (**TFDP**) pipeline for instance segmentation that does not require training on any data in advance. Instance segmentation is sensitive to boundary regions (Tang et al., 2021; Zhang et al., 2021; Li et al., 2023): early in training, predicted masks primarily cover the object's central area, while later stages refine the boundary pixels. Many studies have focused on making models pay more attention to boundaries to enhance final performance (Cheng et al., 2020; Wang et al., 2022a; Borse et al., 2021; Zhang et al., 2021). Therefore, leveraging the rich shape information provided by *pixel-level annotations* of masks, we designed the **Shape Complexity Score** (**SCS**), which characterizes the importance of an instance by calculating the complexity of each mask's boundary. Furthermore, to address the inherent *scale variability* arising from varying object sizes, we implemented **S**cale-**I**nvariant for *SCS (SI-SCS)*, allowing this score to fairly represent the complexity of mask boundaries regardless of size. Additionally, to solve the problem of significant *class imbalances* in instance segmentation tasks (Oksuz et al., 2020), which can lead to noticeable class variability when simply summing instance-level scores to calculate image-level scores. We also design the **C**lass-**B**alanced for *SCS (CB-SCS)*. Specifically, we normalize instance importance scores within each class across images, ensuring each class contributes equally, regardless of its instance count. This design allows us to obtain a unified metric that enables fair comparisons across images, regardless of the number of objects. These effective designs not only address the challenges faced in instance segmentation but also enable the superiority of our method in both time efficiency and performance (see Fig. 1).

As the first work on dataset pruning in instance segmentation and to fully validate the effectiveness of our method as illustrated in Fig. 3, we also adapt existing classification-oriented dataset pruning methods to this task, implementing some baselines with strong performance. Specifically, instead of calculating the importance score for each image in the classification task, we assign an importance score for each pixel based on existing criteria. We then aggregate the scores within each object and subsequently across objects in an image to derive an image-wise score. In our experiments, we conduct extensive comparisons between our implemented baselines and our model-independent TFDP, including performance comparisons, generalizability, and time consumption, to foster a foundational contribution to future work.

In summary, our contributions are as follows:

1) To the best of our knowledge, we are the **first** to introduce a training-free dataset pruning framework for instance segmentation.

2) We adapt existing classification-oriented methods to instance segmentation and establish strong baselines for comparison.

3) We propose a novel model-independent importance criterion **SCS** based on the shape information of masks to prune samples. Additionally, we implement Scale-Invariant and Class-Balanced versions to address the issues of scale variability and class imbalance.

4) Our method achieves the best results on mainstream instance segmentation datasets such as VOC 2012, Cityscapes, and MS COCO, without utilizing any model outputs. It also demonstrates enhanced generalizability across various architectures (including CNN-based and Transformer-based networks) while offering a significantly faster and more practical pruning process.

## 2 RELATED WORK

### 2.1 DATASET COMPRESSION

**Dataset Pruning**, or Coreset Selection, reduces dataset size by selecting key samples based on specific criteria. Herding (Welling, 2009) and Moderate (Xia et al., 2022) measure feature space distances, while Entropy (Coleman et al., 2019) and Cal (Margatina et al., 2021) focus on uncertainty. EL2N (Paul et al., 2021) uses gradient magnitudes to quantify importance. Some methods (Tang et al., 2023; Okanovic et al., 2023; Xu et al., 2023; Dolatabadi et al., 2022) improve training efficiency via online selection. GradMatch (Killamsetty et al., 2021) and Craig (Mirzasoleiman et al., 2020) minimize gradient differences between full and pruned datasets, while ACS (Huang et al., 2023) accelerates quantization-aware training by selecting high-gradient samples. However, these methods primarily focus on classification tasks or NLP task (Nguyen & He, 2025) and have not explored other more complex computer vision tasks further. Additionally, they rely on models to calculate importance scores, which is time-consuming and results in limited generalizability.

**Dataset Distillation** is an another direction to compress datasets, aiming to learn a synthetic dataset that can recover the performance of the full dataset (Nguyen et al., 2021a;b; Zhou et al., 2022; Loo et al., 2022; Zhao et al., 2021; Jiang et al., 2022; Lee et al., 2022; Loo et al., 2023; He et al., 2023; Zhao & Bilen, 2023; Wang et al., 2022b; Zhao et al., 2023; Cazenavette et al., 2022; Du et al., 2023; Cui et al., 2023; Liu et al., 2023; Tukan et al., 2023; Shin et al., 2023; Yin et al., 2023; Yin & Shen, 2023; Shao et al., 2024; Zhou et al., 2024; He et al., 2024; Xiao & He, 2024). However, since every pixel requires a gradient update during the optimization process, the training cost is significantly higher than dataset pruning.

### 2.2 INSTANCE SEGMENTATION

Instance segmentation is a significant and challenging task in computer vision, as it demands both instance-level and pixel-level predictions. The existing methods can be roughly summarized into the following categories. 1) *Top-down methods* (Li et al., 2017; He et al., 2017; Liu et al., 2018b; Huang et al., 2019; Bolya et al., 2019; Chen et al., 2019b; 2020; Zhang et al., 2020) address the problem by approaching it from the object detection perspective, *i.e.*, first detecting objects and then segmenting them within the bounding boxes. Specifically, recent approaches to (Chen et al., 2020; Zhang et al., 2020; Xie et al., 2020) build their methods on the anchor-free object detectors (Tian et al., 2019), showing promising performance. 2) *Bottom-up methods* (Newell et al., 2017; De Brabandere et al., 2017; Liu et al., 2017; Gao et al., 2019) treat the task as label-and-cluster issue, *e.g.*, learning pixel embeddings and clustering them. 3) *Direct methods* (Wang et al., 2020a;b) directly perform instance segmentation without relying on embedding learning or box detection. 4) *Transformer-based methods.* More recently, QueryInst (Fang et al., 2021) and Mask2Former (Cheng et al., 2022) proposed decoding random queries into objects for end-to-end instance segmentation by extending DETR (Carion et al., 2020). Instance segmentation typically requires large training datasets, but training efficiency in this domain remains understudied.

## 3 METHOD

### 3.1 PROBLEM DEFINITION

We first define the task of dataset pruning for instance segmentation. We have a complete training dataset $\mathcal{D} = \{(x_i, y_i)\}_{i=1}^{D}$, where $D$ is the total number of images in the dataset. Here, $x_i \in \mathcal{X}$ is an

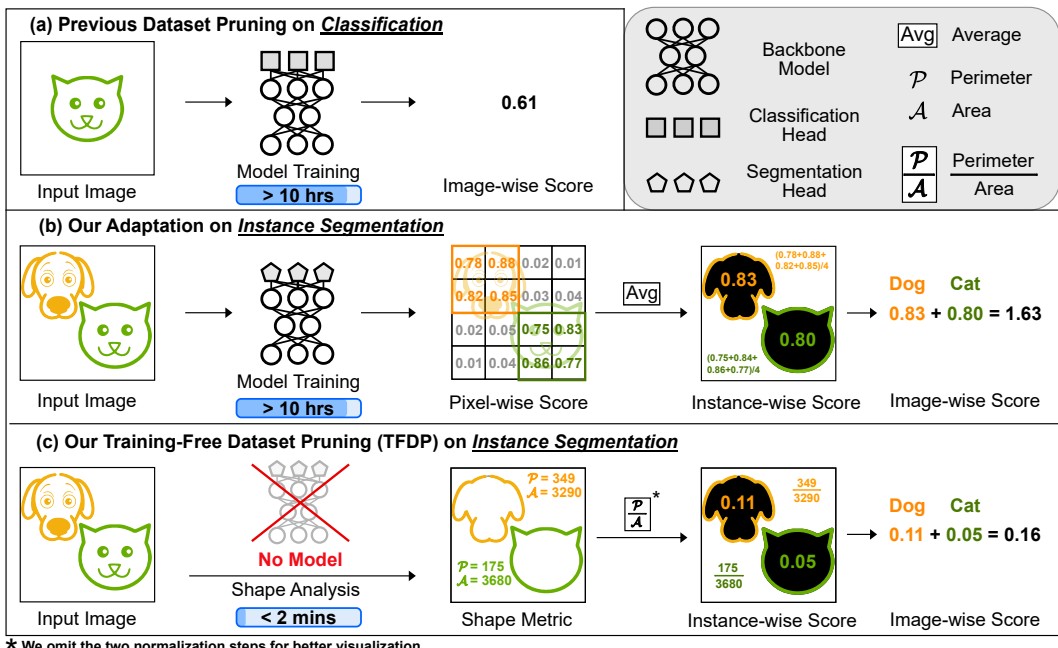

Figure 3: Comparison of different dataset pruning pipelines. (a) Pruning classification dataset requires model training. (b) Our adaptation of previous methods on instance segmentation by training a segemntation head and computing the importance score for each pixel. (c) The proposed method that is training-free and model-independent.

input image, and $y_i \in \mathcal{Y}$ represents the ground truth for instance segmentation. Unlike classification tasks where the ground truth for an image corresponds to a single label (category), the ground truth for instance segmentation assigns a label (category) to each pixel of the image through masks. Specifically, each $y_i$ contains a set of labeled instances:

$$\{(c_{i,j}, m_{i,j})\}_{j=1}^{G_i} \tag{1}$$

where $G_i$ is the number of instances in image $i$, $c_{i,j}$ is the class label of the $j$-th instance, and $m_{i,j}$ is a binary mask that defines the spatial pixel boundaries of the instance.

The objective of dataset pruning is to reduce the size of $\mathcal{D}$ by selecting a subset $\mathbb{S} \subseteq \mathcal{D}$ that maintains or maximizes the performance of a model trained on this reduced set compared to training on the entire dataset $\mathcal{D}$, thereby improving training efficiency and reducing training time. In our approach, we reduce the number of images (image-level) rather than instance annotations (instances-level) since the large volume of image data primarily impacts training time and storage requirements. While annotations for instance segmentation tasks include category labels, bounding boxes, and segmentation masks, their storage footprint remains significantly smaller compared to the images themselves. For instance, in the COCO dataset, images consume about 17 GB, whereas annotations occupy merely 0.7 GB.

## 3.2 ADAPTED BASELINES FOR INSTANCE SEGMENTATION

Existing mainstream dataset pruning methods for classification primarily rely on model-derived logits for each image to calculate scores. However, for instance segmentation, each image requires the model to compute a logit for every pixel rather than for an entire image, making these classification-oriented methods unsuitable without modifications. To ensure a fair comparison, as shown in Fig. 4, we implement some strong baselines: we adapt these above-mentioned methods to suit instance segmentation tasks without altering their criteria for sample selection.

In the context of instance segmentation, we employ the widely-used Mask R-CNN framework as an example, and it can be directly applied to other segmentation models in the same manner. The mask loss in Mask R-CNN is computed using a pixel-wise binary cross-entropy loss between the predicted masks and the ground truth masks for each class. The mask loss $L_m$ for each instance is

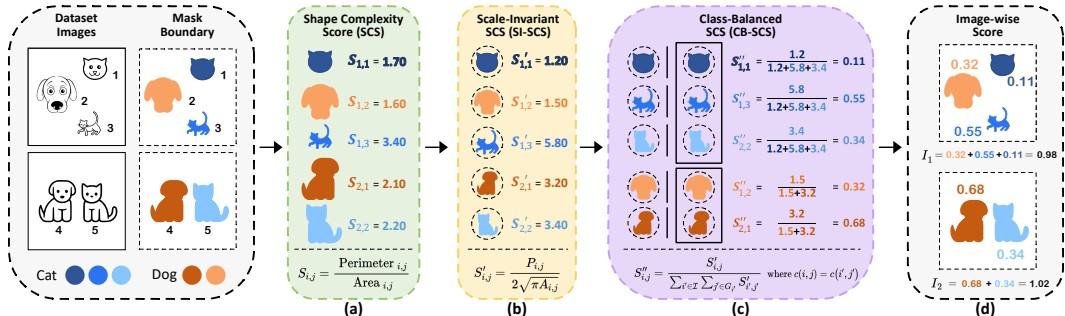

Figure 4: Overview of our proposed framework. We introduce the Shape Complexity Score (*SCS*), in which we leverage the Perimeter-to-Area ratio to represent the boundary complexity. Following this, we apply scale normalization and intra-class normalization to address the inherent scale variability and class imbalance in instance segmentation tasks.

defined as:

$$L_m = -\frac{1}{H \times W} \sum_{i=1}^{H} \sum_{j=1}^{W} \left[ y_{a,b} \log(\hat{y}_{a,b}) + (1 - y_{a,b}) \log(1 - \hat{y}_{a,b}) \right], \tag{2}$$

where $H$ and $W$ are the height and width of the RoI mask respectively, $y_{a,b}$ is the ground truth label at pixel $(a, b)$ (1 for object, 0 for background), and $\hat{y}_{a,b}$ is the predicted logit of the pixel belonging to the object. This formulation allows for pixel-level precision in predicting masks, ensuring detailed and accurate segmentation outputs.

To adapt previous dataset pruning methods to instance segmentation tasks, we extend pixel-level importance scoring to instance-level and image-level scoring. Our model computes per-pixel logits for each predicted instance. We then apply established classification importance metrics (e.g., EL2N) to these logits. The instance-level score is obtained by averaging the pixel scores within each instance. The image's importance score is derived from the sum of its instance scores. Formally, the importance score $I$ for the image $i$ is calculated as follows:

$$I_i = \sum_{j=1}^{G_i} \left( \frac{1}{H_{i,j}} \sum_{a,b} s_{a,b,j} \right), \tag{3}$$

where $G_i$ is the number of instances in the image $i$, $H_{i,j}$ is the number of pixels in the $j$-th instance mask in image $i$, and $s_{a,b,j}$ is the importance score at pixel $(a, b)$ in the $j$-th instance mask. For *pixel-to-instance* score aggregation, we choose *average* to avoid area bias introduced by sum, which favors larger instances with more pixels. For *instance-to-image* aggregation, we use *sum* since images with more instance masks contain more objects and thus more information. Related experiments can also be found in the Appendix D.1.

### 3.3 Training-Free Dataset Pruning for Instance Segmentation

While effective, our adapted baselines are time-consuming and lack cross-architecture generalizability. To overcome these issues, we propose a novel Training-Free Dataset Pruning (TFDP) method for instance segmentation, shown in Fig. 4. The algorithm is outlined in Appendix A.

#### 3.3.1 Shape Complexity Score (SCS)

We propose the **S**hape **C**omplexity **S**core (*SCS*, **Fig. 4a**), a novel model-independent metric that leverages the shape information of instance masks to select challenging samples. For each image $x_i$ in our dataset, we have a set of $G_i$ instance masks $\{M_{i,1}, M_{i,2}, ..., M_{i,G_i}\}$. The *SCS* is calculated for each instance mask as follows:

**1. Mask Preparation:** For each $M_{i,j}$, we obtain a binary mask $B_{i,j}$:

$$B_{i,j}(a,b) = \begin{cases} 1 & \text{if } (a,b) \in M_{i,j} \\ 0 & \text{otherwise} \end{cases} \tag{4}$$

**2. Contour Extraction:** We extract the set of contours $\mathcal{T}_{i,j} = \{T_{i,j,1}, T_{i,j,2}, ..., T_{i,j,Z}\}$ from $B_{i,j}$ and select the primary contour:

$$T_{i,j}^* = \arg \max_{T \in \mathcal{T}_{i,j}} \text{Area}(T) \tag{5}$$

**3. Perimeter Calculation:** The perimeter $P_{i,j}$ is calculated as:

$$P_{i,j} = \text{Perimeter}(T_{i,j}^*) = \sum_{k=1}^{H_{i,j}} \sqrt{(a_k - a_{k+1})^2 + (b_k - b_{k+1})^2}, \tag{6}$$

where $(a_{H_{i,j}+1}, b_{H_{i,j}+1}) = (a_1, b_1)$ to close the contour.

**4. Area Calculation:** The area $A_{i,j}$ of the instance is computed as:

$$A_{i,j} = \text{Area}(T_{i,j}^*) = \frac{1}{2} \left| \sum_{k=1}^{H_{i,j}} (a_k b_{k+1} - a_{k+1} b_k) \right|, \tag{7}$$

where $(a_k, b_k)$ are the coordinates of the $k$-th point in the contour $T_{i,j}^*$ with $H_{i,j}$ points.

**5. Shape Complexity Score Computation:** The *SCS* $S_{i,j}$ for the $j$-th instance in the $i$-th image is defined as:

$$S_{i,j} = \frac{P_{i,j}}{A_{i,j}}. \tag{8}$$

This perimeter-to-area ratio increases with the boundary intricacy of the instance's shape, providing a measure of its complexity. To the best of our knowledge, the *SCS* is the first model-independent metric in the dataset pruning area that leverages the shape information of instances' masks to select challenging samples. This approach effectively alleviates the issue of limited generalizability caused by biases in selection based on specific model predictions and significantly reduces computational overhead.

### 3.3.2 Scale-Invariant SCS (SI-SCS)

The **S**hape **C**omplexity **S**core (*SCS*) exhibits a clear bias towards scale, which substantially impacts instance segmentation tasks, as shown in Fig. 2a. Specifically, smaller-scale instance masks receive higher scores, even when their boundaries are the same. To address this, we propose the **S**cale-**I**nvariant *SCS* (*SI-SCS*, **Fig. 4b**) .

For a polygon with perimeter $P_{i,j}$ and area $A_{i,j}$, scaling by factor $f$ results in:

$$\frac{P'_{i,j}}{A'_{i,j}} = \frac{fP_{i,j}}{f^2 A_{i,j}} = \frac{P_{i,j}}{fA_{i,j}} \tag{9}$$

This decreases with increasing scale, biasing towards smaller instances.

To solve this, we normalize *SCS* using a circle (the shape with the minimum perimeter for a given area) as a reference. Let $S_{i,j}^\circ$ denote the *SCS* of a circle:

$$S_{i,j}^\circ = \frac{P_{i,j}}{A_{i,j}} = \frac{2\pi r}{\pi r^2} = \frac{2}{r} = 2\sqrt{\frac{\pi}{A_{i,j}}} \tag{10}$$

The *SI-SCS* is defined as:

$$S'_{i,j} = \frac{S_{i,j}}{S_{i,j}^\circ} = \frac{P_{i,j}}{2\sqrt{\pi A_{i,j}}} \tag{11}$$

For a scaled polygon (factor $f$), we have:

$$S'_{i,j}(f) = \frac{P_{i,j} \times f}{2\sqrt{\pi \cdot (A_{i,j} \times f^2)}} = \frac{P_{i,j} \times f}{2\sqrt{\pi A_{i,j}} \times f} = \frac{P_{i,j}}{2\sqrt{\pi A_{i,j}}} = S'_{i,j} \tag{12}$$

This demonstrates that *SI-SCS* is scale-invariant, depending only on the boundary complexity and not the instance's scale.

### 3.3.3 CLASS-BALANCED SCS (CB-SCS)

While Scale-Invariance addresses intra-instance shape variability, inter-instance imbalance due to class imbalance remains a significant challenge. Instance segmentation datasets frequently exhibit highly skewed distributions of class instances (Oksuz et al., 2020), as illustrated in Fig. 2a and Appendix B. Naive aggregation of instance-level importance scores within an image can result in rankings disproportionately influenced by high-frequency classes, thereby diminishing the contributions of less frequent classes.

To mitigate this class imbalance issue, we propose the **C**lass-**B**alanced *SCS* (*CB-SCS*, **Fig. 4c**) score to ensures fair representation of all classes, regardless of the number of instances in each class. For each class, we normalize individual instance scores by the total score of all instances in that class across different images. Our normalized scoring method balances class influence, preventing overrepresented classes from dominating while simultaneously preserving the impact of classes with limited examples. Formally, the score for the $i$-th image is,

$$S''_{i,j} = \frac{S'_{i,j}}{\sum_{i' \in \mathcal{I}} \sum_{j' \in G_{i'}} S'_{i',j'}} \quad \text{where } c(i,j) = c(i',j'), \tag{13}$$

where $S'_{i,j}$ is the *SI-SCS* of the $j$-th instance in image $i$, $c(i,j)$ denotes the class of the $j$-th object in image $i$.

**Image-level Score.** Consequently, we can directly sum these normalized scores across all instances in $i$-th image to obtain a balanced image-level score formally defined as $I_i$ (**Fig. 4d**):

$$I_i = \sum_{j}^{G_i} S''_{i,j}, \tag{14}$$

where $G_i$ denotes the number of instances in the image $i$.

## 4 EXPERIMENTS

### 4.1 EXPERIMENT SETTINGS

**Datasets.** To evaluate the proposed method TFDP, we conduct instance segmentation experiments on three mainstream datasets VOC 2012 (Everingham et al., 2010), Cityscapes (Cordts et al., 2016), and MS COCO (Lin et al., 2014).

**Evaluation Metrics.** For all datasets, we evaluate instance segmentation results using the standard COCO protocol and report the average precision metrics for both mask and box AP (averaged over IoU thresholds): mAP, $AP_{50}$, $AP_{75}$, $AP_S$, $AP_M$, $AP_L$. Due to the page limits more details about datasets and evaluation metrics are provided in Appendix C.1 and C.2, respectively.

**Baseline Comparisons.** Six baselines are used for comparison: 1) **Random**; 2) **Entropy** (Coleman et al., 2019); 3) **Forgetting** (Toneva et al., 2019); 4) **EL2N** (Paul et al., 2021); 5) **AUM** (Pleiss et al., 2020); 6) **CCS** (Zheng et al., 2023). For all methods (except Random), we make specific adaptations for the instance segmentation task, as described in Sec. 3.2. As is common practice (He et al., 2017; Wang et al., 2020b), all network backbones are pre-trained on the ImageNet-1k classification set (Deng et al., 2009) and then fine-tuned on the instance segmentation dataset. For hyperparameters, we follow the settings described in the original paper with details provided in the Appendix C.3.

### 4.2 PRIMARY RESULTS

**MS COCO (Tab. 1).** Tab. 1a shows the results on the COCO dataset. Our TFDP method outperforms all baselines across all pruning rate settings. For example, when pruning 50% of samples, TFDP still achieves 53.4% $AP_{50}$, which is 1.7% and 2.2% higher than adapted Entropy and EL2N, respectively. Moreover, TFDP's performance with only 80% of the data already matches the performance with 100% of the data. Additionally, with just 50% of the data selected by our TFDP method, it consistently outperforms random pruning by 30%. Impressively, compared to other baselines that require model training, the selection time for the model-independent TFDP is almost negligible. For

| $p$ | Time | mAP | | | | | AP$_{50}$ | | | | | AP$_{75}$ | | | | |
|---|---|---|---|---|---|---|---|---|---|---|---|---|---|---|---|---|
| | | 0% | 20% | 30% | 40% | 50% | 0% | 20% | 30% | 40% | 50% | 0% | 20% | 30% | 40% | 50% |
| Random | - | 34.2 | 33.6 | 32.1 | 31.1 | 30.8 | 55.2 | 54.5 | 52.8 | 51.1 | 51.0 | 36.5 | 35.6 | 34.1 | 33.2 | 32.7 |
| Forgetting | 20.29 h | - | 33.1 | 32.3 | 31.4 | 30.4 | - | 54.2 | 53.4 | 52.2 | 51.2 | - | 35.2 | 34.3 | 33.4 | 32.1 |
| Entropy | 21.16 h | - | 33.2 | 32.3 | 31.4 | 30.9 | - | 54.4 | 53.5 | 52.5 | 51.7 | - | 35.5 | 34.5 | 33.2 | 32.6 |
| EL2N | 12.37 h | - | 33.4 | 32.1 | 31.2 | 30.5 | - | 54.5 | 52.9 | 51.7 | 51.2 | - | 35.6 | 34.2 | 33.2 | 32.0 |
| AUM | 20.29 h | - | 33.5 | 32.4 | 31.5 | 31.0 | - | 54.6 | 53.3 | 52.4 | 51.7 | - | 35.5 | 34.7 | 33.4 | 32.8 |
| CCS | 20.29 h | - | 33.4 | 32.4 | 31.7 | 31.5 | - | 54.1 | 53.3 | 52.6 | 52.3 | - | 35.6 | 34.4 | 33.6 | 33.2 |
| Ours | **0.014 h** | - | **34.4** | **33.6** | **33.1** | **32.5** | - | **55.5** | **54.8** | **54.2** | **53.4** | - | **36.7** | **35.4** | **35.1** | **34.3** |
| Diff. | ↑ **1349×** | - | **+0.8** | **+1.5** | **+2.0** | **+1.7** | - | **+1.0** | **+2.0** | **+3.1** | **+2.4** | - | **+1.1** | **+1.3** | **+1.9** | **+1.6** |

(a) The mask AP (%) results compare different dataset pruning baselines on COCO.

| $p$ | Time | mAP$^{bb}$ | | | | | AP$_{50}$$^{bb}$ | | | | | AP$_{75}$$^{bb}$ | | | | |
|---|---|---|---|---|---|---|---|---|---|---|---|---|---|---|---|---|
| | | 0% | 20% | 30% | 40% | 50% | 0% | 20% | 30% | 40% | 50% | 0% | 20% | 30% | 40% | 50% |
| Random | - | 37.7 | 37.0 | 35.3 | 34.0 | 33.8 | 58.3 | 57.6 | 56.1 | 53.8 | 54.3 | 41.1 | 40.1 | 38.0 | 37.3 | 36.3 |
| Forgetting | 20.29 h | - | 36.8 | 35.6 | 34.5 | 34.0 | - | 57.7 | 56.2 | 55.2 | 54.4 | - | 40.4 | 38.7 | 37.5 | 36.8 |
| Entropy | 21.16 h | - | 36.7 | 35.8 | 34.7 | 34.3 | - | 57.6 | 56.7 | 55.8 | 55.2 | - | 40.0 | 39.2 | 37.8 | 37.4 |
| EL2N | 12.37 h | - | 36.9 | 35.7 | 34.7 | 34.0 | - | 57.7 | 56.4 | 55.1 | 54.5 | - | 40.1 | 38.9 | 37.6 | 36.5 |
| AUM | 20.29 h | - | 37.0 | 35.8 | 34.8 | 34.3 | - | 57.9 | 56.6 | 55.6 | 55.1 | - | 40.6 | 39.0 | 38.1 | 37.1 |
| CCS | 20.29 h | - | 36.8 | 35.7 | 35.2 | 34.7 | - | 57.6 | 56.5 | 56.1 | 55.7 | - | 40.3 | 39.1 | 38.2 | 37.5 |
| Ours | **0.014 h** | - | **37.8** | **37.2** | **36.7** | **35.9** | - | **58.8** | **58.1** | **57.6** | **56.9** | - | **41.1** | **40.2** | **39.9** | **38.8** |
| Diff. | ↑ **1349×** | - | **+0.8** | **+1.9** | **+2.7** | **+2.1** | - | **+1.2** | **+2.0** | **+3.8** | **+2.6** | - | **+1.0** | **+2.2** | **+2.6** | **+2.5** |

(b) The bounding-box (bb) AP (%) results compare different dataset pruning baselines on COCO.

Table 1: Results on COCO with backbone network Mask R-CNN. The pruning rate $p$ represents the percentage of data removed from the full training dataset during pruning. The performance on the full dataset is indicated by $p = 0\%$. `Time` indicates the time consumption for sample ranking, with details provided in Sec. 4.3. `Diff` for the improvement in time represents the average improvement, excluding Random since it does not consume time. `Diff` for the performance denotes the difference between our TFDP method and random pruning, as all baselines are also our adapted methods.

| Dataset | VOC | | | | | | Cityscapes | | | | | |
|---|---|---|---|---|---|---|---|---|---|---|---|---|
| $p$ | Time | 0% | 20% | 30% | 40% | 50% | Time | 0% | 20% | 30% | 40% | 50% |
| Random | - | 40.9 | 39.4 | 32.0 | 29.0 | 23.7 | - | 27.6 | 26.1 | 21.8 | 19.0 | 16.9 |
| Forgetting | 21.21 min | - | 33.6 | 30.8 | 28.1 | 21.6 | 5.54 h | - | 25.8 | 23.2 | 19.3 | 17.1 |
| Entropy | 21.75 min | - | 38.4 | 34.2 | 31.7 | 29.3 | 5.61 h | - | 26.4 | 22.2 | 20.1 | 17.2 |
| El2N | 12.70 min | - | 39.1 | 35.3 | 32.1 | 29.8 | 3.01 h | - | 26.2 | 22.6 | 20.3 | 17.4 |
| AUM | 21.21 min | - | 35.2 | 31.0 | 26.3 | 19.2 | 5.54 h | - | 25.3 | 24.5 | 21.2 | 18.4 |
| CCS | 21.21 min | - | 38.8 | 35.4 | 34.3 | 30.8 | 5.54 h | - | 25.4 | 24.1 | 19.9 | 17.0 |
| Ours | **0.12 min** | - | **40.3** | **38.6** | **36.2** | **33.4** | **0.0051 h** | - | **27.5** | **25.4** | **23.4** | **19.4** |
| Diff. | ↑ **164×** | - | **+0.9** | **+6.6** | **+7.2** | **+9.7** | ↑ **100×** | - | **+1.4** | **+3.6** | **+4.4** | **+2.5** |

Table 2: The mask AP (%) results compare different dataset pruning baselines on VOC and Cityscapes. The pruning rate $p$ represents the percentage of data removed from the full training dataset during pruning. The performance on the full dataset is indicated by $p = 0\%$.

| Model | SOLO-v2 | | QueryInst | |
|---|---|---|---|---|
| $p$ | 40% | 50% | 40% | 50% |
| Random | 51.4 | 51.1 | 53.3 | 52.8 |
| Entropy | 52.8 | 51.6 | 55.6 | 53.9 |
| EL2N | 52.1 | 50.3 | 55.0 | 52.7 |
| AUM | 52.5 | 51.0 | 55.6 | 54.0 |
| CCS | 53.0 | 52.1 | 55.0 | 53.5 |
| Ours | **53.1** | **52.3** | **55.9** | **55.0** |
| Diff. | **+1.7** | **+1.2** | **+2.6** | **+2.2** |

Table 3: The AP$_{50}$ (%) results in the generalization ability to different architectures on COCO dataset.

more results on the COCO dataset, please refer to the Appendix D.2. **Bounding-Box (bb) AP Results.** Following previous work (He et al., 2017), we also report the bounding-box object detection results on three datasets in Tab. 1b. Our method TFDP significantly outperforms random pruning at all pruning rate settings. Notably, by selecting only 50% of the data, our TFDP consistently surpasses all baseline methods that select 70% of the data (30% pruning rate) in both mAP$^{bb}$ and AP$_{50}$$^{bb}$ performance. Compared to the Mask AP results, the performance improvement in bounding-box AP is more pronounced.

**VOC and Cityscapes (Tab. 2).** Performance results on VOC and Cityscapes are reported in Tab. 2. Our TFDP method demonstrates superior performance on VOC. For instance, with the pruning rate of 40% and 50%, Mask R-CNN trained on the pruned VOC achieves mask AP$_{50}$ accuracies of 36.2% and 33.4%, surpassing random pruning by 7.2% and 9.7%, respectively. On Cityscapes, TFDP also consistently shows improvements. For example, at a pruning rate of 40%, TFDP exceeds the performance of random pruning at 40% by 4.4%, and at 30% by 1.3%. Additionally, our TFDP

| SI | CB | VOC | | | Cityscapes | | | COCO | | |
|---|---|---|---|---|---|---|---|---|---|---|
| p | | 30% | 40% | 50% | 30% | 40% | 50% | 30% | 40% | 50% |
| random | | 32.0 | 29.0 | 23.7 | 22.1 | 19.0 | 16.9 | 53.8 | 52.1 | 52.0 |
| - | - | 36.6 | 34.0 | 28.8 | 22.4 | 21.5 | 19.1 | 54.2 | 52.9 | 52.5 |
| ✓ | - | 37.8 | 35.4 | 32.4 | 22.2 | 22.8 | 17.9 | 54.2 | 53.3 | 52.1 |
| - | ✓ | 35.9 | 33.5 | 30.7 | 24.9 | 22.4 | 17.1 | 54.4 | 54.0 | 52.7 |
| ✓ | ✓ | **38.6** | **36.2** | **33.4** | **25.4** | **23.4** | **19.4** | **54.8** | **54.2** | **53.4** |

Table 4: Ablation study of two components **SI-SCS** (SI) and **CB-SCS** (CB). When neither normalization is used (both marked as '-'), it indicates that only **SCS** is applied.

| | mAP | | | | | AP$_{50}$ | | | | |
|---|---|---|---|---|---|---|---|---|---|---|
| | 0% | 20% | 30% | 40% | 50% | 0% | 20% | 30% | 40% | 50% |
| Random | 36.4 | 35.4 | 35.0 | 35.6 | 33.8 | 61.8 | 60.5 | 60.8 | 60.6 | 58.9 |
| Entropy | - | 34.7 | 35.6 | 34.5 | 34.3 | - | 61.3 | 62.6 | 61.4 | 60.3 |
| EL2N | - | 34.2 | 34.1 | 35.2 | 32.1 | - | 59.2 | 59.7 | 61.8 | 57.3 |
| AUM | - | 36.3 | 34.9 | 34.4 | 33.9 | - | 62.6 | 60.9 | 59.4 | 59.4 |
| CCS | - | 36.1 | 36.1 | 35.0 | 34.0 | - | 61.7 | 61.7 | 60.7 | 59.7 |
| Ours | - | **36.9** | **36.6** | **36.6** | **36.6** | - | **62.8** | **63.9** | **63.4** | **62.8** |
| Diff. | - | +1.5 | +1.6 | +1.0 | +2.8 | - | +2.3 | +3.1 | +2.8 | +3.9 |

Table 5: The mask AP (%) results on Cityscapes (**pretrained on COCO**).

method consistently outperforms the adapted SOTA methods (e.g., EL2N and CCS) on VOC and Cityscapes at all pruning rate settings. For more results on the VOC and Cityscapes dataset, please refer to the Appendix D.3.

**Generalization to Other CNN-based and Transformer-based Models (Tab. 3).** In Tab. 3, we show an experiment assessing the effectiveness of images selected by Mask R-CNN when tested on instance segmentation networks with new architectures. We included other CNN-based network SOLO-v2 (Wang et al., 2020b), as well as Transformer-based networks QueryInst (Fang et al., 2021) to test for generalizability. Appendix D.4 shows more results of the generalization evaluation. Additionally, we conducted experiments on different backbones (e.g., ResNet-101 and ResNeXt-101) as shown in Appendix D.5, further validating the scalability of our method.

**Ablation Study (Tab. 4).** Tab. 4 demonstrates the effectiveness of our components on three datasets. We notice using them alone does not give satisfactory results, and applying both **SI-SCS** and **CB-SCS** delivers the best performance. Further analysis on the Scale-Invariance design is provided in Appendix D.6.

## 4.3 MORE ANALYSIS

**Robust Performance Across Training Paradigms (Tab. 5).** Our TFDP is a method that prepares an pruned dataset independent of any specific training process. To comprehensively evaluate its effectiveness, we use TFDP-pruned datasets in two scenarios: 1) Models pre-trained on ImageNet-1k (Results shown in Tab. 1, Tab. 2 and Tab. 3), 2) Models pre-trained on COCO (He et al., 2017; Liu et al., 2018a) as shown in Tab. 5. In both cases, models are fine-tuned on either the full or TFDP-pruned dataset. Impressively, TFDP outperforms all baselines across different pruning rates in both cases. Even when pruning 50% of the data, it (AP$_{50}$ 62.8%) surpasses the performance achieved with the full dataset (AP$_{50}$ 61.8%). This demonstrates TFDP's effectiveness as a universal preprocessing step regardless of the subsequent training strategy. More experimental settings and results for training scenarios 2) are provided in Appendix D.7.

**Performance Gain at High Pruning Rates (Fig. 5).** To further validate our method, we also conduct experiments under high pruning rates, as shown in Fig. 5. Compared to random pruning, our method consistently improves performance across pruning rates from 60% to 90%.

**Time Consumption Results (Tab. 6).** We detail the time required to calculate importance scores for all samples in the dataset, including model training, score computation via model inference, and stratified selection. All time tests were conducted in the same environment, with details available in Appendix D.8, and the times reported are averages of three trials. As shown in Tab. 6, our method significantly reduces sample selection time, with greater time-saving effects on larger datasets, highlighting its effectiveness in the big data era.

**Visualization (Fig. 6).** Fig. 6 shows that our method effectively distinguishes "hard" (top-ranked) and "easy" images (least-ranked). Top-ranked images contain either many intricate-shaped objects (subfig-a, b, c) or instances of rare classes (subfig-d, e), while least-ranked images typically contain a single object with a simple contour (subfig-f, g, h, i, j, k). We use the top-1 image (subfig-d, e) from MS COCO to explain how scale variability and class imbalance issues are addressed. First, despite having various scales, **SI-SCS** assigns all toasters similar scores, illustrating the effectiveness of our **S**cale-**I**nvariant **SCS** (**SI-SCS**). Second, despite having relatively simple shapes, **CB-SCS** assigns

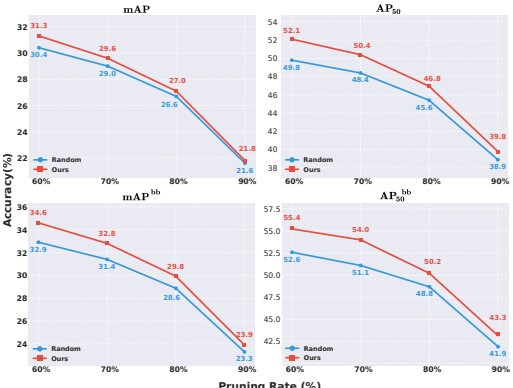

Figure 5: Experiments of high pruning rates (from 60% to 90%) on MS COCO dataset. Segmentation metrics include mAP, $AP_{50}$ and the corresponding bounding-box version, $mAP^{bb}$ and $AP_{50}^{bb}$.

| Dataset | | | Ours | EL2N | Entropy | AUM/Forgetting | CCS |
|---|---|---|---|---|---|---|---|
| VOC | $\mathcal{T}$ | | - | 722.49 s | 1265.40 s | 1272.31 s | 1272.31 s |
| | $\mathcal{I}$ | | 7.19 s | 39.39 s | 39.71 s | - | - |
| | $\mathcal{S}$ | | - | - | - | - | 0.009 s |
| | **Total** | | **7.19 s** | 761.88 s | 1305.11 s | 1272.31 s | 1272.32 s |
| Cityscapes | $\mathcal{T}$ | | - | 2.89 h | 5.49 h | 5.54 h | 5.54 h |
| | $\mathcal{I}$ | | 182.4 s | 429.29 s | 432.33 s | - | - |
| | $\mathcal{S}$ | | - | - | - | - | 0.029 s |
| | **Total** | | **182.4 s** | 3.01 h | 5.61 h | 5.54 h | 5.54 h |
| COCO | $\mathcal{T}$ | | - | 11.35 h | 20.13 h | 20.29 h | 20.29 h |
| | $\mathcal{I}$ | | 50.4 s | 3676.48 s | 3696.69 s | - | - |
| | $\mathcal{S}$ | | - | - | - | - | 0.99 s |
| | **Total** | | **50.4 s** | 12.37 h | 21.16 h | 20.29 h | 20.29 h |

Table 6: Detailed time calculation. $\mathcal{T}$ denotes the model training time. $\mathcal{I}$ is the score computation via model inference. $\mathcal{S}$ is the stratified selection time, for CCS (Zheng et al., 2023).

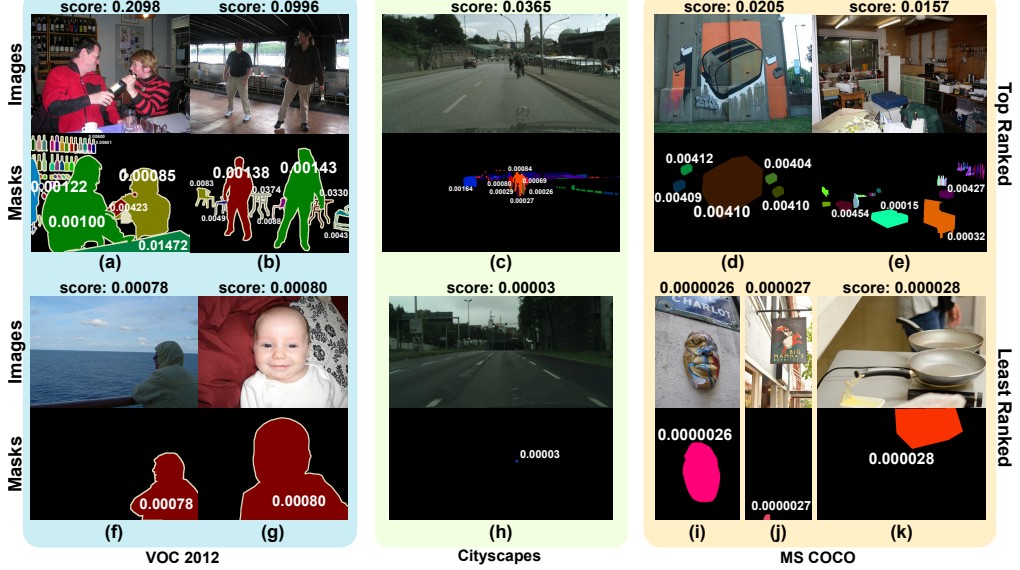

Figure 6: Visualization of TFDP-selected images on different datasets. The original aspect ratios of the images are preserved. Scores of too small objects are omitted for better visualization.

a high score to this image due to the rarity of the class "toaster", showcasing our **C**lass-**B**alanced **SCS (CB-SCS)**. The distribution Fig. 8 in the Appendix B reveals that the number of toasters is the second-to-last least counted object. More visualizations, examples and analysis are in Appendix D.9.

## 5  CONCLUSION AND FUTURE WORK

We introduce Training-Free Dataset Pruning (TFDP) for instance segmentation, leveraging mask annotations and novel scoring methods to address scale variability and class imbalance. Our approach significantly reduces pruning time while improving performance, outperforming adapted state-of-the-art baselines and demonstrating strong cross-architecture generalization. In future work, we plan to extend the training-free concept to diverse datasets. This expansion will include video dataset pruning by incorporating temporal information and motion patterns, language dataset pruning through analysis of text structure, syntax, and semantic richness. Exploring the theoretical foundations of training-free pruning methods is another interesting direction.

## 6 ACKNOWLEDGMENT

This work was supported in part by A*STAR Career Development Fund (CDF) under C243512011, in part by A*STAR Centre for Frontier AI Research (CFAR) and in part by CFAR Internship Award and Research Excellence (CIARE).

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

## A  TRAINING-FREE DATASET PRUNING (TFDP) ALGORITHM

Algorithm 1 illustrates the code implementation process of our TFDP.

---

**Algorithm 1** Training-Free Dataset Pruning (TFDP)

---

**Input:** Complete training dataset $\mathcal{D}$. Number of images to select $K$.

 1: **Initialize:** $\mathbb{S} = \emptyset$, set of selected samples
 2: **for** $i = 1$ to $D$ **do**
 3:     **for** $j = 1$ to $G_i$ **do**
 4:         Calculate $P_{i,j}$ and $A_{i,j}$ for each mask $M_{i,k}$ in image $i$
 5:         Compute SCS $S_{i,j}$                                          ▷ Defined in Equation 8
 6:         Scale-Invariant SCS by circle ratio $S'_{i,j}$                ▷ Defined in Equation 12
 7:     **end for**
 8: **end for**
 9: **for** $i = 1$ to $D$ **do**
10:     **for** $k = 1$ to $G_i$ **do**
11:         Compute Class-Balanced SCS $S''_{i,j}$ for each instance mask $m_{i,j}$ in image $i$
12:     **end for**
13:     Calculate image score $I_i$                                       ▷ Defined in Equation 14
14: **end for**
15: Sort all images in $\mathcal{D}$ based on $I_i$ in descending order
16: Select top $K$ images based on sorted scores to form subset $S$
**Output:** Selected subset $\mathbb{S} = \{(x_i, y_i)\}_{i=1}^{K}$; Importance scores based on $I_i$

---

# B DATASETS ANALYSIS

We show statistical details of more datasets, including VOC, Cityscapes and COCO. As shown in Fig. 8, there are significant differences in object areas across the datasets, along with a clear class imbalance.

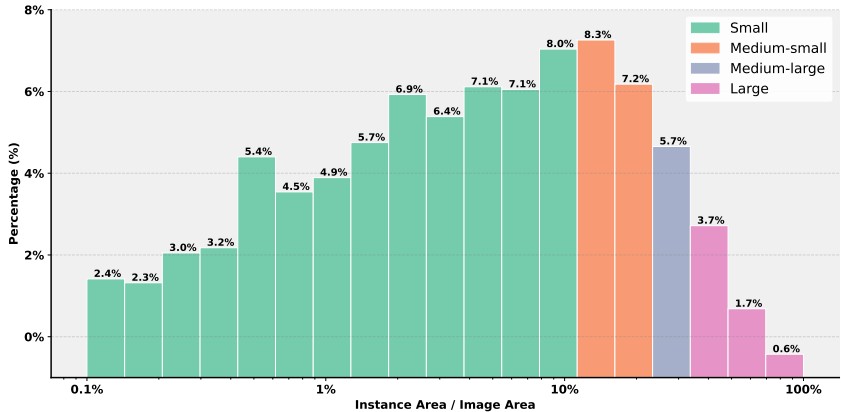

(a) Distribution of VOC 2012 dataset.

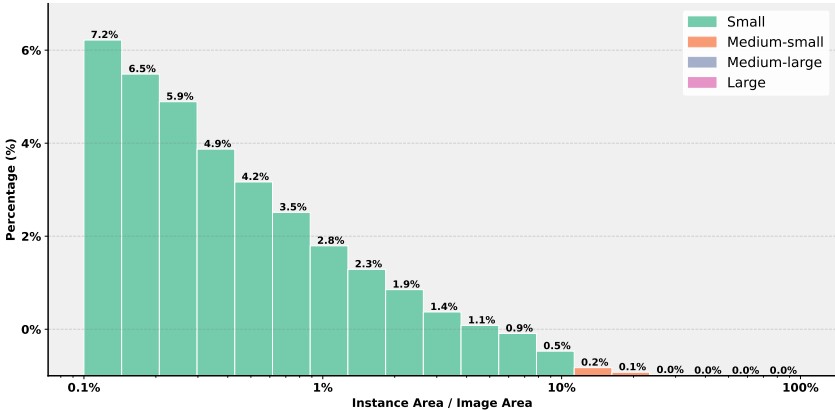

(b) Distribution of Cityscapes dataset.

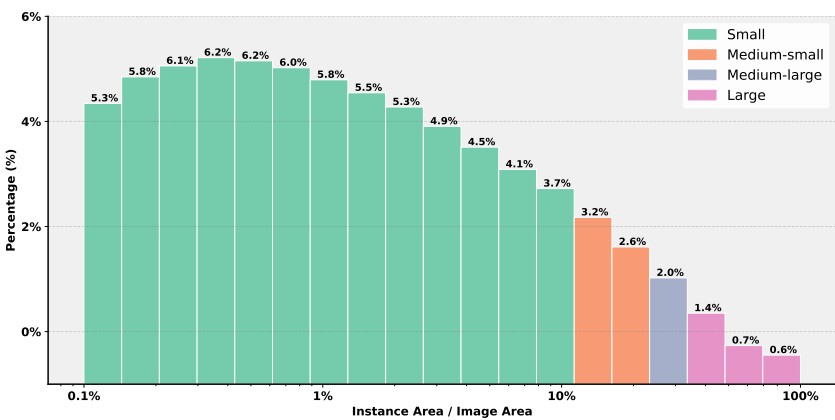

(c) Distribution of MS COCO dataset.

Figure 7: Distribution of area of instances.

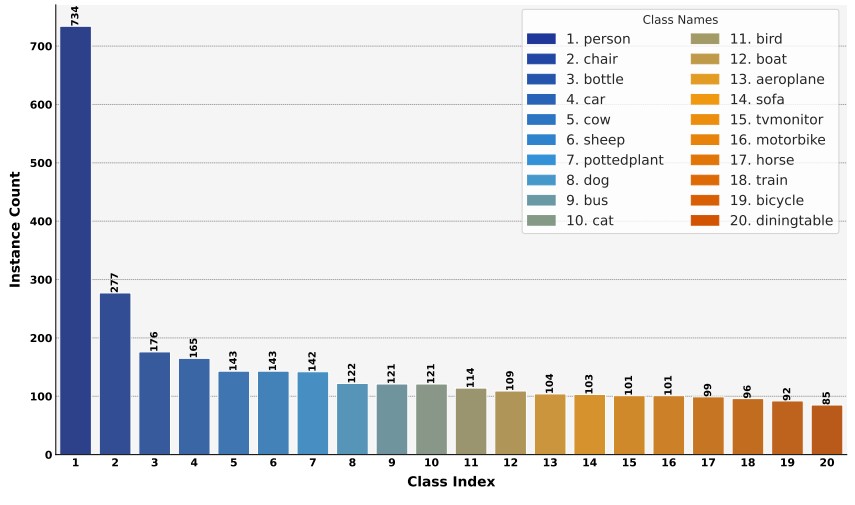

(a) Distribution of VOC 2012 dataset.

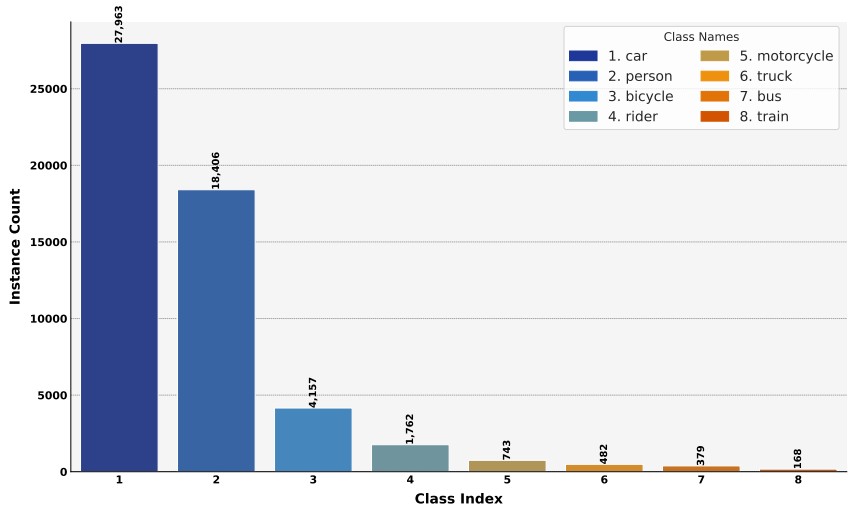

(b) Distribution of Cityscapes dataset.

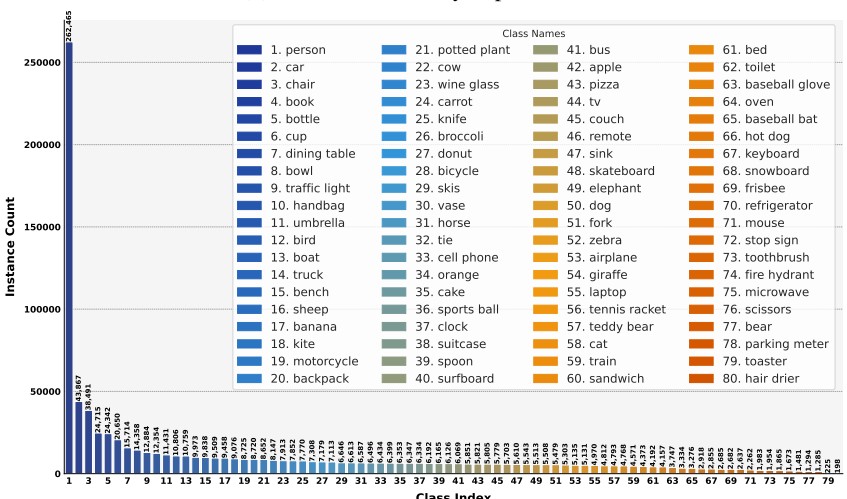

(c) Distribution of MS COCO dataset.

Figure 8: Distribution of number of instances.

## C  EXPERIMENT DETAILS

### C.1  DATASET DETAILS

Our experiments are conducted on three different datasets:

- **Pascal VOC 2012** features 2,913 images, split into 1,464 for training and 1,449 for validation, with 6,929 segmentations available for instance segmentation task.
- **Cityscapes** includes 9 object categories designed for instance-level semantic labeling. This dataset is more complex as each image may contain a significantly higher number of instances per class compared to VOC, most of which are quite small. It contains 2975 training images collected from 18 cities and 500 validation images from 3 different cities.
- **MS COCO** is a popular instance segmentation dataset, which contains an 80-category label set with instance-level annotations. Following previous works (He et al., 2017; Wang et al., 2020b), we use the COCO `train 2017` (118K training images) for training, and the ablation study is carried out on the `val 2017` (5K validation images).

### C.2  EVALUATION METRICS

For all datasets, we evaluate instance segmentation results using the standard COCO protocol (Lin et al., 2014). This protocol provides a comprehensive set of metrics to assess the performance of object detection and instance segmentation models. We report the average precision (AP) metrics for both mask and bounding box predictions, which are averaged over multiple Intersection over Union (IoU) thresholds. The key metrics we report are:

- **mAP (mean Average Precision):** This is the primary metric, calculated by averaging AP across all object categories and IoU thresholds (typically from 0.5 to 0.95 in steps of 0.05).
- **$AP_{50}$:** Average Precision at IoU threshold of 0.5. This metric is more lenient, considering detections as correct if they have at least 50% overlap with the ground truth.
- **$AP_{75}$:** Average Precision at IoU threshold of 0.75. This is a stricter metric, requiring more accurate localization of objects.
- **$AP_S$, $AP_M$, $AP_L$:** These metrics evaluate performance on small, medium, and large objects respectively.
  - $AP_S$: AP for small objects (area $< 32^2$ pixels)
  - $AP_M$: AP for medium objects ($32^2 <$ area $< 96^2$ pixels)
  - $AP_L$: AP for large objects (area $> 96^2$ pixels)

These size-specific metrics help assess the model's performance across different scales of objects, which is crucial for understanding its effectiveness in various real-world scenarios.

### C.3  EXPERIMENTS SETTINGS

Except for the generalization and scalability experiments, we use ResNet-50 as the backbone for Mask R-CNN and FPN to extract multi-scale features. All models and training hyperparameters were trained using the default hyperparameters as specified in MMDetection (Chen et al., 2019a).

**Adapted Baseline Settings.** For EL2N, in alignment with the description of the model in the early training stage from the original paper, we use the model at halfway through the total training duration to compute EL2N scores. For CCS, we strictly follow the hyperparameters in the original paper, including the hard pruning rate $\beta$ and the number of strata $k$. It is worth noting that our TFDP does not contain any hyperparameters, further demonstrating the practicality of our TFDP approach.

**Primary Experiment Settings.** For the VOC experiment, we used a single NVIDIA 3090 GPU. For the Cityscapes experiment, we used two NVIDIA 3090 GPUs. For the COCO experiment, we used two NVIDIA A100 80G GPUs. **Time Consumption Experiment Settings.** To ensure a fair comparison, all time consumption experiments were conducted on the same machine: PyTorch on Ubuntu 20.04, with NVIDIA RTX 3090 GPUs and CUDA 11.3. We used two NVIDIA 3090 GPUs for training and one NVIDIA 3090 GPU for inference.

# D MORE EXPERIMENTS

## D.1 COMPARISON BETWEEN SUMMATION AND AVERAGE

We compared two methods for aggregating object scores through pixels: summation and average. As shown in Tab. 7, the average-based method performs significantly better than the summation-based method. This is because the summation-based method tends to favor larger masks, i.e., instances with more pixels, causing smaller instances to be overlooked.

| | mAP | | | $AP_{50}$ | | | $AP_{75}$ | | | $AP_S$ | | | $AP_M$ | | | $AP_L$ | | |
|---|---|---|---|---|---|---|---|---|---|---|---|---|---|---|---|---|---|---|
| | Avg. | Sum | Diff. | Avg. | Sum | Diff. | Avg. | Sum | Diff. | Avg. | Sum | Diff. | Avg. | Sum | Diff. | Avg. | Sum | Diff. |
| Forgetting | 28.7 | 31.0 | +2.3 | 47.7 | 52.1 | +4.4 | 31.1 | 35.1 | +4.0 | 12.3 | 16.7 | +4.4 | 31.6 | 33.4 | +1.8 | 40.1 | 42.8 | +2.7 |
| Entropy | 28.1 | 31.1 | +3.0 | 47.0 | 51.7 | +4.7 | 30.6 | 34.7 | +4.1 | 12.7 | 16.2 | +3.5 | 31.1 | 33.2 | +2.1 | 39.9 | 42.7 | +2.8 |
| EL2N | 25.4 | 30.3 | +4.9 | 45.3 | 50.9 | +5.6 | 28.9 | 33.2 | +4.3 | 11.0 | 14.3 | +3.3 | 30.1 | 32.1 | +2.0 | 38.7 | 41.3 | +2.6 |
| AUM | 26.7 | 29.9 | +3.2 | 46.6 | 51.2 | +4.6 | 29.7 | 33.9 | +4.2 | 11.4 | 15.6 | +4.2 | 30.3 | 32.9 | +2.6 | 39.0 | 41.2 | +2.2 |
| CCS | 28.4 | 30.9 | +2.5 | 47.2 | 51.8 | +4.6 | 31.0 | 34.8 | +3.8 | 12.1 | 16.7 | +4.6 | 31.1 | 33.2 | +2.1 | 40.3 | 42.9 | +2.6 |

Table 7: Comparison between averaging (`Avg.`) and summation (`Sum`).

## D.2 MS COCO RESULT ON $AP_S$, $AP_M$, AND $AP_L$

Tab. 8 shows more detailed AP results on the COCO dataset for different object areas (small, medium, large). Our TFDP consistently achieves stable improvements.

| | $AP_S$ | | | | | $AP_M$ | | | | | $AP_L$ | | | | |
|---|---|---|---|---|---|---|---|---|---|---|---|---|---|---|---|
| | 0% | 20% | 30% | 40% | 50% | 0% | 20% | 30% | 40% | 50% | 0% | 20% | 30% | 40% | 50% |
| Random | 16.1 | 15.9 | 14.1 | 13.1 | 13.2 | 36.7 | 36.0 | 34.6 | 33.2 | 33.2 | 49.9 | 48.8 | 48.1 | 46.2 | 45.6 |
| Forgetting | - | 15.6 | 15.0 | 14.6 | 14.5 | - | 35.7 | 35.1 | 34.2 | 33.7 | - | 49.1 | 46.5 | 44.9 | 43.6 |
| Entropy | - | 15.5 | 14.9 | 14.3 | 14.4 | - | 35.9 | 35.0 | 34.1 | 33.9 | - | 48.9 | 46.7 | 45.1 | 43.8 |
| EL2N | - | 15.6 | 14.5 | 14.4 | 14.4 | - | 36.3 | 35.2 | 34.1 | 33.6 | - | 47.9 | 46.3 | 44.6 | 43.2 |
| AUM | - | 15.8 | 14.7 | 14.6 | 14.4 | - | 36.3 | 35.4 | 34.3 | 34.1 | - | 48.5 | 46.6 | 45.3 | 44.0 |
| CCS | - | 15.8 | 15.2 | 14.8 | 14.7 | - | 36.0 | 34.9 | 34.6 | 34.1 | - | 48.4 | 46.7 | 45.0 | 45.2 |
| Ours | - | **16.2** | **15.5** | **15.5** | **15.1** | - | **37.1** | **36.3** | **36.0** | **35.3** | - | **49.3** | **48.5** | **46.9** | **46.2** |
| Diff. | - | **+0.3** | **+1.4** | **+2.4** | **+1.9** | - | **+1.1** | **+1.7** | **+2.8** | **+2.1** | - | **+0.5** | **+0.4** | **+0.7** | **+0.6** |

(a) The mask AP (%) results for different object areas (small, medium, large) compare different dataset pruning baselines on COCO.

| | $AP_S^{bb}$ | | | | | $AP_M^{bb}$ | | | | | $AP_L^{bb}$ | | | | |
|---|---|---|---|---|---|---|---|---|---|---|---|---|---|---|---|
| | 0% | 20% | 30% | 40% | 50% | 0% | 20% | 30% | 40% | 50% | 0% | 20% | 30% | 40% | 50% |
| Random | 21.9 | 21.8 | 19.6 | 18.4 | 18.4 | 40.9 | 40.2 | 38.4 | 37.1 | 36.8 | 48.9 | 47.8 | 46.4 | 44.1 | 44.3 |
| Forgetting | - | 20.8 | 20.7 | 20.1 | 20.1 | - | 40.1 | 39.9 | 38.4 | 37.8 | - | 47.5 | 45.3 | 43.7 | 42.4 |
| Entropy | - | 21.5 | 20.7 | 20.3 | **20.7** | - | 40.3 | 39.5 | 38.3 | 37.9 | - | 47.6 | 46.0 | 43.8 | 43.3 |
| EL2N | - | 21.1 | 20.8 | 20.2 | 19.8 | - | 40.7 | 39.7 | 38.4 | 37.9 | - | 47.1 | 45.1 | 43.5 | 41.8 |
| AUM | - | 22.0 | 20.7 | 20.4 | 20.2 | - | 40.8 | 40.0 | 38.7 | 38.4 | - | 47.7 | 45.7 | 44.0 | 43.0 |
| CCS | - | 21.6 | 20.8 | 20.6 | 20.0 | - | 40.4 | 39.5 | 38.9 | 38.2 | - | 47.4 | 45.6 | 44.3 | 43.9 |
| Ours | - | **22.4** | **21.1** | **21.4** | 20.6 | - | **41.1** | **40.4** | **40.5** | **39.5** | - | **48.3** | **47.7** | **46.0** | **45.3** |
| Diff. | - | **+0.6** | **+1.5** | **+3.0** | **+2.2** | - | **+0.9** | **+2.0** | **+3.4** | **+2.7** | - | **+0.5** | **+1.3** | **+1.9** | **+1.0** |

(b) The bbox AP (%) results for different object areas (small, medium, large) compare different dataset pruning baselines on COCO.

Table 8: More results on COCO. The pruning rate $p$ represents the percentage of data removed from the full training dataset during pruning. The performance on the full dataset is indicated by $p = 0\%$. `Diff.` denotes the difference between our method and random pruning, and the improvement in time is the average improvement.

## D.3 CITYSCAPES RESULT ON MAP, AP50, AND AP75

Tab. 9 shows more detailed AP results on the Cityscapes dataset for different IoU thresholds. Our TFDP consistently achieves stable improvements.

| | mAP | | | | | AP$_{50}$ | | | | | AP$_{75}$ | | | | |
|---|---|---|---|---|---|---|---|---|---|---|---|---|---|---|---|
| | 0% | 20% | 30% | 40% | 50% | 0% | 20% | 30% | 40% | 50% | 0% | 20% | 30% | 40% | 50% |
| Random | 12.5 | 11.2 | 9.1 | 8.0 | 7.2 | 27.6 | 26.1 | 21.8 | 19.0 | 16.9 | 10.0 | 8.7 | 6.7 | 6.1 | 5.3 |
| Forgetting | - | 11.3 | 10.3 | 8.3 | 7.3 | - | 25.8 | 23.2 | 19.3 | 17.1 | - | 8.2 | 7.6 | 5.7 | 4.9 |
| Entropy | - | 11.5 | 10.2 | 8.4 | 7.0 | - | 26.4 | 23.8 | 20.7 | 17.2 | - | 8.6 | 7.4 | 5.8 | 5.1 |
| EL2N | - | 11.6 | 10.1 | 9.2 | **8.0** | - | 26.2 | 23.1 | 20.8 | 17.9 | - | 8.9 | 7.7 | 7.4 | **6.6** |
| AUM | - | 11.1 | 10.7 | 9.2 | 7.8 | - | 25.3 | 24.5 | 21.2 | 18.4 | - | 8.1 | 8.1 | 7.0 | 6.2 |
| CCS | - | 10.9 | 10.4 | 8.5 | 7.0 | - | 25.4 | 24.0 | 20.0 | 17.0 | - | 8.2 | 8.0 | 6.8 | 5.0 |
| Ours | - | **12.4** | **10.9** | **9.6** | 7.5 | - | **27.5** | **25.4** | **23.3** | **19.4** | - | **9.5** | **8.2** | **7.2** | 5.4 |
| Diff. | - | **+1.2** | **+1.8** | **+1.6** | **+0.3** | - | **+1.4** | **+3.6** | **+4.3** | **+2.5** | - | **+0.8** | **+1.5** | **+1.1** | **+0.1** |

(a) The mask AP (%) results for different IoU thresholds (0.5 to 0.95, 50, 75) compare different dataset pruning baselines on Cityscapes.

| | mAP$^{bb}$ | | | | | AP$_{50}{}^{bb}$ | | | | | AP$_{75}{}^{bb}$ | | | | |
|---|---|---|---|---|---|---|---|---|---|---|---|---|---|---|---|
| | 0% | 20% | 30% | 40% | 50% | 0% | 20% | 30% | 40% | 50% | 0% | 20% | 30% | 40% | 50% |
| Random | 15.7 | 14.5 | 12.1 | 10.6 | 9.3 | 33.0 | 31.7 | 27.2 | 24.2 | 21.7 | 12.3 | 11.2 | 8.7 | 7.7 | 6.7 |
| Forgetting | - | 13.9 | 13.1 | 11.0 | 9.7 | - | 31.0 | 29.3 | 26.4 | 22.0 | - | 11.4 | 9.9 | 7.4 | 6.6 |
| Entropy | - | 14.8 | 13.4 | 11.5 | 9.4 | - | 31.6 | 29.6 | 26.5 | 21.6 | - | 11.8 | 10.1 | 8.0 | 6.8 |
| EL2N | - | 14.9 | 13.3 | 11.7 | **10.4** | - | 32.0 | 29.2 | 26.0 | 23.0 | - | 11.8 | 9.8 | 9.1 | 7.2 |
| AUM | - | 14.3 | 13.2 | 12.0 | 10.3 | - | 31.0 | 29.9 | 26.5 | **23.8** | - | 11.1 | 10.0 | 9.0 | **7.3** |
| CCS | - | 13.5 | 13.3 | 10.9 | 9.4 | - | 30.1 | 29.7 | 25.3 | 21.6 | - | 10.0 | 9.6 | 7.8 | 6.7 |
| Ours | - | **15.3** | **14.1** | **12.6** | 10.0 | - | **32.4** | **31.2** | **28.5** | 23.5 | - | **11.3** | **10.1** | **9.1** | 6.6 |
| Diff. | - | **+0.8** | **+2.0** | **+2.0** | **+0.7** | - | **+0.7** | **+4.0** | **+4.3** | **+1.8** | - | **+0.1** | **+1.4** | **+1.4** | **-0.1** |

(b) The bbox AP (%) results for different IoU thresholds (0.5 to 0.95, 50, 75) compare different dataset pruning baselines on Cityscapes.

Table 9: More results on Cityscapes. The pruning rate $p$ represents the percentage of data removed from the full training dataset during pruning. The performance on the full dataset is indicated by $p = 0\%$. `Diff.` denotes the difference between our method and random pruning, and the improvement in time is the average improvement.

## D.4 CROSS-ARCHITECTURE GENERALIZATION EXPERIMENTS

Tab. 10 shows more detailed results on models with different architectures (such as SOLO-v2 and QueryInst) for different IoU thresholds, and our TFDP consistently achieves stable improvements. Since the adapted baselines rely on model-specific pruning, the data selected using Mask R-CNN shows varying degrees of degradation when applied to other architectures. This is especially evident in the Transformer-based model QueryInst, where the data selected by CNN-based Mask R-CNN performs worse than random selection under many pruning rate settings.

| | SOLO-v2 | | | | | | QueryInst | | | | | |
|---|---|---|---|---|---|---|---|---|---|---|---|---|
| $p = 50\%$ | mAP | AP$_{50}$ | AP$_{75}$ | AP$_S$ | AP$_M$ | AP$_L$ | mAP | AP$_{50}$ | AP$_{75}$ | AP$_S$ | AP$_M$ | AP$_L$ |
| Random | 31.3 | 51.1 | 32.6 | 11.6 | 34.2 | 48.4 | 33.3 | 52.8 | 35.6 | 15.0 | 35.4 | 51.8 |
| Entropy | 31.4 | 51.6 | 33.0 | 12.5 | 35.0 | 46.6 | 33.5 | 53.9 | 35.6 | 16.1 | 36.3 | 49.4 |
| EL2N | 30.7 | 50.3 | 32.2 | 11.5 | 34.6 | 45.3 | 32.8 | 52.7 | 35.0 | 15.7 | 36.0 | 48.4 |
| AUM | 31.0 | 51.0 | 32.3 | 11.5 | 34.7 | 45.6 | 33.9 | 54.0 | 36.4 | 15.8 | 36.8 | 49.9 |
| CCS | 31.8 | 52.1 | 33.2 | 12.1 | 35.5 | 47.7 | 33.3 | 53.5 | 35.6 | 15.6 | 35.9 | 49.8 |
| Ours | **32.2** | **52.3** | **34.1** | **12.3** | **35.5** | **48.6** | **34.5** | **55.0** | **36.9** | **15.8** | **37.5** | **51.9** |
| Diff. | **+0.9** | **+1.2** | **+1.5** | **+0.7** | **+1.3** | **+0.2** | **+1.2** | **+2.2** | **+1.3** | **+0.8** | **+2.1** | **+0.1** |

Table 10: More detailed results in the generalization ability to different architectures on COCO dataset.

## D.5 NETWORK SCALING EXPERIMENTS

Tab. 11 shows results to verify TFDP's scalability on the COCO dataset for different backbones (such as ResNet-101 and ResNeXt-101). On backbones with more parameters and stronger performance, our TFDP can still further improve the model's performance, demonstrating the good scalability of our method.

| Validation Network | | mAP | | | | $AP_{50}$ | | | | $AP_{50}$ | | | |
|---|---|---|---|---|---|---|---|---|---|---|---|---|---|
| | | 20% | 30% | 40% | 50% | 20% | 30% | 40% | 50% | 20% | 30% | 40% | 50% |
| ResNet-101 | Random | 35.5 | 34.8 | 34.0 | 33.1 | 56.6 | 55.7 | 54.7 | 53.8 | 37.9 | 36.9 | 36.3 | 35.0 |
| | Ours | **35.8** | **35.3** | **34.9** | **34.3** | **57.1** | **56.6** | **56.4** | **55.6** | **38.2** | **37.6** | **37.3** | **36.6** |
| | Diff. | +0.3 | +0.5 | +0.9 | +1.2 | +0.5 | +0.9 | +1.7 | +1.8 | +0.3 | +0.7 | +1.0 | +1.6 |
| ResNeXt-101 | Random | 36.7 | 36.2 | 35.4 | 34.3 | 58.4 | 58.0 | 56.8 | 55.4 | 39.2 | 38.4 | 37.8 | 36.3 |
| | Ours | **37.2** | **36.6** | **36.1** | **35.5** | **59.2** | **58.8** | **58.2** | **57.6** | **39.9** | **39.0** | **38.4** | **37.9** |
| | Diff. | +0.5 | +0.4 | +0.7 | +1.2 | +0.8 | +0.8 | +1.4 | +2.2 | +0.7 | +0.6 | +0.6 | +1.6 |

(a) The mask AP (%) results for different IoU thresholds (0.5 to 0.95, 50, 75) of different backbones on COCO.

| Validation Network | | mAP | | | | $AP_{50}^{bb}$ | | | | $AP_{50}^{bb}$ | | | |
|---|---|---|---|---|---|---|---|---|---|---|---|---|---|
| | | 20% | 30% | 40% | 50% | 20% | 30% | 40% | 50% | 20% | 30% | 40% | 50% |
| ResNet-101 | Random | 39.2 | 38.4 | 37.6 | 36.4 | 59.7 | 58.8 | 57.9 | 56.8 | 42.8 | 41.9 | 40.8 | 39.6 |
| | Ours | **39.8** | **39.4** | **38.9** | **38.1** | **60.4** | **60.2** | **59.7** | **58.9** | **43.5** | **42.6** | **42.4** | **41.2** |
| | Diff. | +0.6 | +1.0 | +1.3 | +1.7 | +0.7 | +1.4 | +1.8 | +2.1 | +0.7 | +0.7 | +1.6 | +1.6 |
| ResNeXt-101 | Random | 40.8 | 40.3 | 39.1 | 38.0 | 61.6 | 61.4 | 59.9 | 58.7 | 44.7 | 44.2 | 42.8 | 41.5 |
| | Ours | **41.4** | **41.0** | **40.3** | **39.6** | **62.6** | **62.1** | **61.4** | **60.9** | **45.4** | **45.0** | **44.0** | **43.4** |
| | Diff. | +0.6 | +0.7 | +1.2 | +1.6 | +1.0 | +0.7 | +1.5 | +2.2 | +0.7 | +0.8 | +1.2 | +1.9 |

(b) The bbox AP (%) results for different IoU thresholds (0.5 to 0.95, 50, 75) of different backbones on COCO.

Table 11: The $AP_{50}$ (%) results in the scalability ability to the different backbones of Mask R-CNN on the COCO dataset.

## D.6 ANALYSIS ON SCALE-INVARIANCE DESIGN

To further validate the effect of SI design, which is applied to mitigate the small-scale bias of the SCS, we compare the Average Precision scores at different scales ($AP_S$, $AP_M$, $AP_L$) following the COCO standard metrics before and after applying SI design in Tab. 12. Our results indicate that, prior to applying SI, the SCS scores were significantly biased towards smaller-scale images, resulting in poor performance in $AP_M$ and $AP_L$. Conversely, after applying the Scale-Invariant design, there was a noticeable improvement in $AP_L$ levels, without any decline in $AP_S$ results.

| $p$ | 30% | | | 40% | | | 50% | | |
|---|---|---|---|---|---|---|---|---|---|
| | $AP_S$ | $AP_M$ | $AP_L$ | $AP_S$ | $AP_M$ | $AP_L$ | $AP_S$ | $AP_M$ | $AP_L$ |
| w/o SI | 20.6 | 12.9 | 17.8 | 20.5 | 12.2 | 16.3 | 19.5 | 11.3 | 13.8 |
| w/ SI | 22.6 | 13.4 | 20.7 | 21.7 | 14.1 | 18.3 | 20.2 | 12.5 | 15.9 |
| ↑ | +2.0 | +0.5 | **+2.9** | +1.2 | +1.9 | **+2.0** | +0.7 | +1.2 | **+2.1** |

Table 12: Ablation study of the SI-SCS. Comparison of mask AP for objects of different scales followed by COCO official metrics (Lin et al., 2014).

## D.7 MORE RESULTS ON THE SECOND TRAINING PARADIGM

As mentioned in He et al. (2017): A major difficulty with the Cityscapes dataset is the limited number of training samples available for certain categories, such as *truck, bus* and *train*, which only have around 200-500 examples each, making it challenging to train models effectively. To partially remedy this issue, we follow He et al. (2017) and report the results using COCO pre-training to verify our method further. Specifically, we strictly follow He et al. (2017) and use a pre-trained COCO Mask R-CNN model to initialize 7 categories in Cityscapes, while the *rider* category is randomly initialized.

In this fine-tuning setting, as shown in Tab. 13, our TFDP achieves more significant improvements. Specifically, our method consistently surpasses random pruning as well as other baselines. Notably, our pruning is better than the full dataset, even at a 50% pruning ratio. This may be due to TFDP pruning some low-quality or noisy data that could negatively impact model performance. Furthermore, in the fine-tuning setting, using the full dataset might lead to overfitting, thereby affecting the

final performance. These results demonstrate that our method is not only applicable but also more promising under the fine-tuning setting.

| | mAP | | | | | AP$_{50}$ | | | | |
|---|---|---|---|---|---|---|---|---|---|---|
| | 0% | 20% | 30% | 40% | 50% | 0% | 20% | 30% | 40% | 50% |
| Random | 36.4 | 35.4 | 35.0 | 35.6 | 33.8 | 61.8 | 60.5 | 60.8 | 60.6 | 58.9 |
| Entropy | - | 34.7 | 35.6 | 34.5 | 34.3 | - | 61.3 | 62.6 | 61.4 | 60.3 |
| EL2N | - | 34.2 | 34.1 | 35.2 | 32.1 | - | 59.2 | 59.7 | 61.8 | 57.3 |
| AUM | - | 36.3 | 34.9 | 34.4 | 33.9 | - | 62.6 | 60.9 | 59.4 | 59.4 |
| CCS | - | 36.1 | 36.1 | 35.0 | 34.0 | - | 61.7 | 61.7 | 60.7 | 59.7 |
| Ours | - | **36.9** | **36.6** | **36.6** | **36.6** | - | **62.8** | **63.9** | **63.4** | **62.8** |
| Diff. | - | **+1.5** | **+1.6** | **+1.0** | **+2.8** | - | **+2.3** | **+3.1** | **+2.8** | **+3.9** |

(a) The mask AP (%) results compare different dataset pruning baselines on Cityscapes (pre-trained on COCO).

| | mAP$^{bb}$ | | | | | AP$_{50}{}^{bb}$ | | | | |
|---|---|---|---|---|---|---|---|---|---|---|
| | 0% | 20% | 30% | 40% | 50% | 0% | 20% | 30% | 40% | 50% |
| Random | 40.9 | 39.6 | 39.6 | 40.2 | 39.5 | 66.3 | 64.9 | 64.9 | 65.2 | 64.0 |
| Entropy | - | 39.5 | 41.0 | 39.2 | 39.4 | - | 63.9 | 67.3 | 66.1 | 65.4 |
| EL2N | - | 38.1 | 39.3 | 35.2 | 37.2 | - | 62.7 | 65.1 | 61.8 | 62.5 |
| AUM | - | 36.3 | 40.8 | 39.8 | 39.0 | - | 62.6 | 65.8 | 64.2 | 64.0 |
| CCS | - | 41.0 | 41.0 | 40.1 | 38.7 | - | 66.0 | 66.0 | 64.8 | 64.3 |
| Ours | - | **42.1** | **41.1** | **41.4** | **42.0** | - | **67.4** | **68.7** | **67.5** | **67.9** |
| Diff. | - | **+2.5** | **+1.5** | **+1.2** | **+2.5** | - | **+2.5** | **+3.8** | **+2.3** | **+3.9** |

(b) The bbox AP (%) results compare different dataset pruning baselines on Cityscapes (pre-trained on COCO).

Table 13: Fine-tuned results with COCO pre-trained Mask R-CNN on Cityscapes.

## D.8 TIME CONSUMPTION DETAILS

To ensure a fair comparison, all time consumption experiments were conducted on a same machine: PyTorch on Ubuntu 20.04, with NVIDIA RTX 3090 GPUs and CUDA 11.3. We used two NVIDIA 3090 GPUs for training and one NVIDIA 3090 GPU for inference.

The following section provides explanations of some time consumption details. EL2N (Paul et al., 2021) and Entropy (Coleman et al., 2019) measure the importance of samples by calculating the distance between output logits and the ground-truth one-hot labels, which includes the time for both model training and inference (Scoring). According to the original settings of EL2N, it only requires model weights in the early training stage, hence the training time is shorter than Entropy. AUM (Pleiss et al., 2020) and Forgetting (Toneva et al., 2019) assess the difficulty of samples by tracking changes in the logits across different epochs during training, thus the process includes model training and scoring through training logs. The time difference between the two is minimal, and we report them as a single method. For CCS, in addition to the time taken to obtain AUM scores, the time for Stratified selection should also be factored in. However, the proposed TFDP does not require a model and only needs the time for scoring through pixel-level annotations, which is significantly less than the time required by existing model-based sample selection methods.

## D.9 MORE VISUALIZATIONS

To further explore the effectiveness of our method, we provide more visualization on VOC, Cityscapes, and COCO in Fig. 9. Notably, *SCS* itself can successfully distinguish "hard" and "easy" images as shown in Fig 9 (a). We can visually tell the top-ranked images are more complex than the least-ranked images. By applying **S**cale-**I**nvariant *SCS* , the effect of scale has been eliminated. We can observe from the least-ranked images in Fig. 9 (b) that the criterion changes from simply picking the largest objects to easier shapes (e.g., circles). Lastly, by considering class balance, the criterion selects not only the complex shapes but also the images that can best cover the distribution of the dataset.

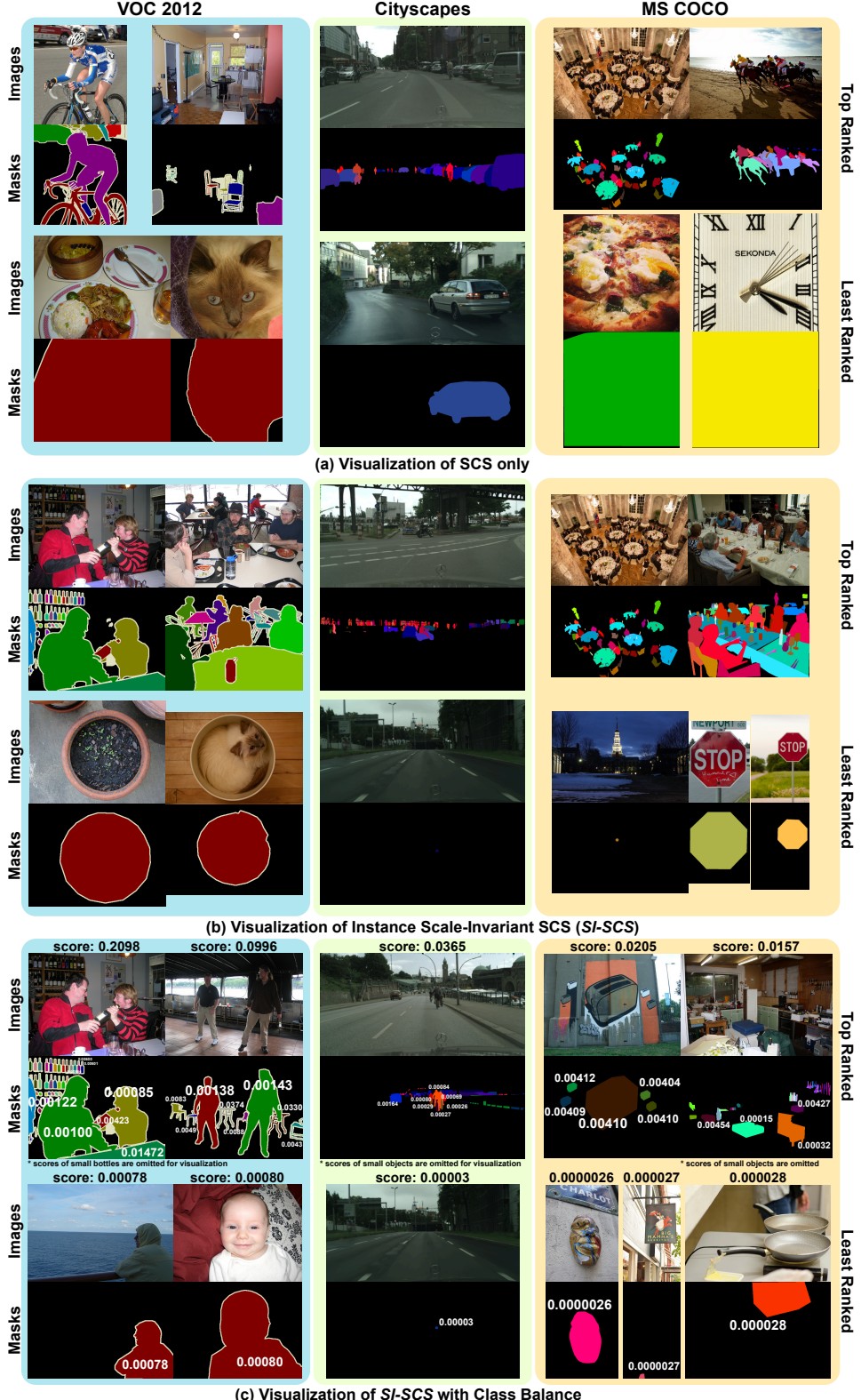

Figure 9: More visualization of the proposed method.

