# OpenReview forum: "Training-Free Dataset Pruning for Instance Segmentation"
_ICLR.cc/2025/Conference — ICLR 2025 Poster_

### Official Review · Reviewer_p9Ls · 2024-10-28

**Soundness:** 3
**Presentation:** 3
**Contribution:** 3
**Rating:** 6
**Confidence:** 4

**Summary:**

The cropping method for classification tasks is ineffective in instance segmentation applications. The paper proposes a Training-Free Dataset Pruning (TFDP) method, which utilizes the shape and class information in image annotations to design a Shape Complexity Score (SCS) to address issues of instance region variation and class imbalance.

**Strengths:**

This paper proposes the Shape Complexity Score method around the simple P/A ratio concept. It introduces a relatively simple cropping method in the instance segmentation task. On the other hand, it significantly shortens the time for sample sorting in the cropping method. In addition, this method is more prominent in solving the instance segmentation task compared to previous methods. As mentioned above, the proposed method in this work has certain innovation, the paper framework is logically clear, and the integrity of the paper is relatively high.

**Weaknesses:**

1.It is suggested that the author add literature on different pruning methods in the instance segmentation task in the Introduction section or the Dataset Pruning section of the related work, or provide a relevant summary analysis.

2.It is suggested that detailed explanations be provided for formulas (5) to (12) in the paper, such as what does T represent in formula (5)?

3.The discussion and derivation of SI-SCS content in Section 3.3.2 of the text are not detailed enough, and it is recommended to add supplementary explanations.

4.The experimental part of the paper is not sufficiently comprehensive. 1) Important details regarding the training and inference stages are not provided in the experimental setup; 2) No detailed explanation is given for the poorer results in Table 4 after using SI. In addition, the authors have not provided a detailed analysis of the experimental results for all figures and tables in the experimental section.

5.The visualization results presented in Figures 6 and 9 can be intuitively observed to show that for some complex scenarios, when dealing with targets of more intricate shapes, or when addressing the local areas of the targets, the performance may be somewhat underwhelming.

**Questions:**

1. Combining theoretical and experimental results, we have considerable doubts about the use of P/A ratio and other metrics as Shape Complexity Scores in the instance-level score calculation in the paper.

2. In SCS, the CB-SCS actually balances the number of instance objects in each diagram, but for datasets like COCO, the number of instances in some categories is very small. Does this potentially cause bias in the data distribution? Additionally, is the consumption of computing resources relatively high?

3. For the ablation experiments of SI and CB in SCS, it can be seen from Table 4 that the experimental results obtained by increasing SI or CB alone are not satisfactory. However, the authors did not provide relevant analysis in the paper. Why does increasing the module individually affect the performance of the original model? Why does the combination of the two result in better outcomes?

---

> ### Author Response · Authors · 2024-11-24
> **Rebuttal 1: W1, W2, W3, W4(1), W4(2), Q3**
>
> Thank you for your valuable feedback. We are greatly encouraged by your praise for our work as "being innovative", "logically clear" and "high integrity”.
>
> The concerns and questions you pointed out have provided valuable insights. We have addressed the issues you mentioned and present our responses below. ${\color{blue}\text{Added changes are highlighted in blue in the revision.}}$
>
> >W1:  It is suggested that the author add literature on different pruning methods in the instance segmentation task in the Introduction section or the Dataset Pruning section of the related work, or provide a relevant summary analysis.
>
> Thank you for your suggestions.
>
> 1.As the **first** dataset pruning method for instance segmentation, our approach pioneers this task. To our knowledge, there have been no new related works about dataset pruning in instance segmentation to date.
>
> 2.Following your suggestions, to make the citations in this paper more rigorous, we have added references to the latest articles on other related tasks, such as object detection [a, b] and visual instruct tuning [c, d].
>
> [a] Lee, et al. Coreset Selection for Object Detection, CVPR Workshop (2024)
>
> [b] Qi, et al. Fetch and Forge: Efficient Dataset Condensation for Object Detection, NeurIPS (2024)
>
> [c] Lee, et al.  Concept-skill Transferability-based Data Selection for Large Vision-Language Models, EMNLP (2024)
>
> [d] LESS: Selecting Influential Data for Targeted Instruction Tuning, ICML (2024)
>
> > W2: It is suggested that detailed explanations be provided for formulas (5) to (12) in the paper, such as what does T represent in formula (5)?
>
> Thank you for your insightful feedback. We apologize for any confusion caused by our unclear expressions. All the symbols related to equations (5) through (12) in Sections 3.3.1 to 3.3.2 are defined in detail in Sections 3.1 and 3.2.
>
> To fully address your questions, we have added a more detailed explanation of formulas (5) to (12) in the new revision (${\color{blue}\text{Appendix Sec E.1}}$ More explanation about the formula), including the representation of $T$ in formula (5) as you mentioned.
>
> Finally, we appreciate the chance to clarify these points. Our method reduces computational overhead by using shape information to prune less complex instances, thereby boosting training efficiency without compromising performance. We hope this explanation resolves the concerns raised and clearly conveys our method.
>
>
> > W3: The discussion and derivation of SI-SCS content in Section 3.3.2 of the text are not detailed enough, and it is recommended to add supplementary explanations.
>
> Thank you for your comments regarding the derivation of the Scale-Invariant Shape Complexity Score (SI-SCS). We have added a more detailed explanation and justification for SI-SCS in our revision, highlighted in ${\color{blue}\text{blue}}$ (${\color{blue}\text{Appendix Sec E.2}}$: More discussion and derivation of SI-SCS).
>
> These additions clarify how SI-SCS operates independently of scale, ensuring consistent assessment of shape complexity. We value your feedback and the opportunity to enhance our work.
>
>
> > W4(1): The experimental part of the paper is not sufficiently comprehensive. 1) Important details regarding the training and inference stages are not provided in the experimental setup.
>
> Thank you for your attention to the experimental details of our paper. In Section C of the appendix, we provide detailed descriptions of **Dataset Details (C.1)**, **Evaluation Metrics (C.2)**, and **Experiment Settings (C.3)**. Section C.3 includes various details such as **General Settings**, **Adapted Baseline Settings**, **Primary Experiment Settings**, and **Time Consumption Experiment Settings**. if you require further clarification, please let us know which points require additional details, and we will include the relevant information in the supplementary material for the next revision.
>
> > W4(2): Explanation for poorer results in Table 4 after using SI.
>
> > Q3: For the ablation experiments of SI and CB in SCS, it can be seen from Table 4 that the experimental results obtained by increasing SI or CB alone are not satisfactory. However, the authors did not provide relevant analysis in the paper. Why does increasing the module individually affect the performance of the original model? Why does the combination of the two result in better outcomes?
>
> To fully address your concerns, we have supplemented the detailed analysis and explanation in the new revision, highlighted in ${\color{blue}\text{blue}}$ (${\color{blue}\text{Appendix Sec E.3}}$: More analysis of the experimental results, Table 4: Ablation Study.). In summary, while SI is essential for improving overall performance, CB addresses specific limitations by mitigating class imbalances, especially in datasets like Cityscapes. Acting as complementary approaches, the combination of SI and CB delivers the best performance across diverse datasets.

---

> ### Author Response · Authors · 2024-11-24
> **Rebuttal 1: W4(3), W5, Q1, Q2**
>
> > W4(3): In addition, the authors have not provided a detailed analysis of the experimental results for all figures and tables in the experimental section.
>
> Thank you for your suggestions. In the main body of the paper and the Appendix, we have presented extensive experimental results and provided detailed qualitative and quantitative analyses demonstrating the advantages of our method compared to other baselines.
>
> To fully address your concerns, we have supplemented the explanations for all experimental results in the new revision, highlighted in ${\color{blue}\text{blue}}$ (${\color{blue}\text{Appendix Sec E.3}}$: More analysis of the experimental results). We hope the additional analyses adequately address your questions. If further clarification is needed, please let us know which points require additional explanation, and we will include the relevant information in the supplementary material for the next revision.
>
>
> >W5: The visualization results presented in Figures 6 and 9 can be intuitively observed to show that for some complex scenarios, when dealing with targets of more intricate shapes, or when addressing the local areas of the targets, the performance may be somewhat underwhelming.
>
> Thank you for your comments. We would like to clarify that the scores depicted in the figures are importance scores for training prioritization, not direct measures of model performance. Higher scores indicate that a sample is crucial and should be retained, particularly in complex scenes, to enhance model training. Lower scores suggest that a sample is less important and can be pruned. This method focuses training on samples that improve the model's ability to handle diverse scenarios.
>
> We apologize for the confusion caused by our initial presentation. We will clarify the role of the visualized scores more clearly in the new revision.
>
> >Q1: Combining theoretical and experimental results, we have considerable doubts about the use of P/A ratio and other metrics as Shape Complexity Scores in the instance-level score calculation in the paper.
>
> Thank you for your comments and for highlighting concerns regarding our use of the perimeter-to-area (P-to-A) ratio as a measure of shape complexity. We have conducted extensive analyses in the new revision, highlighted in ${\color{blue}\text{blue}}$ (${\color{blue}\text{Appendix Sec E.4}}$: More details about Shape Complexity Score (SCS)).
>
> The perimeter-to-area (P-to-A) ratio is not only theoretically sound but also empirically effective in distinguishing between simple and complex shapes in a consistent and unbiased manner. This metric is particularly advantageous in our context of dataset pruning, where identifying and focusing on more complex instances can lead to more robust and generalized model training.
>
> We hope this explanation addresses the concerns raised and supports the validity of our approach. We appreciate the opportunity to clarify this aspect of our research and are confident that the P-to-A ratio is a reliable measure of shape complexity, enhancing our dataset pruning method.
>
>
> >Q2: In SCS, the CB-SCS actually balances the number of instance objects in each diagram, but for datasets like COCO, the number of instances in some categories is very small. Does this potentially cause bias in the data distribution? Additionally, is the consumption of computing resources relatively high?
>
> Thank you for your questions concerning the potential for data bias. We appreciate the opportunity to clarify these aspects of our CB-SCS method.
>
> 1.**Class-Balanced (CB) Design.**
> 1) The CB-SCS method is specifically designed to mitigate the issue of class imbalance, a prevalent problem in instance segmentation datasets that can disproportionately favor majority classes at the expense of minority ones. By normalizing instance scores within each class, CB-SCS ensures that no single class dominates the learning process.
> 2) This approach seeks to encourage more equitable contributions from all classes, regardless of their frequency in the dataset, thereby promoting a more balanced model training process. Consequently, CB-SCS has the potential to enhance the model’s robustness and generalizability, improving its performance across diverse and imbalanced class distributions.
>
> 2.**Computational Resources.**
> 1) The implementation of CB-SCS is computationally efficient, adding negligible overhead to the overall process. The normalization step involves a straightforward calculation of the ratio of individual scores to the total scores within each class. With a computational **complexity of O(n)**, this operation is performed once per dataset and cached for subsequent use.
> 2) In our test, the execution of the CB module was measured to take approximately 3.1 seconds on COCO, a minimal duration compared to the overall pruning process (50.4 seconds), where the primary time consumption is attributed to calculating the SCS.

---

> ### Comment · Reviewer_p9Ls · 2024-11-24
> **Response to authors**
>
> Thanks for the feedback. My new question is as follow:
>
> 1. In your response to "Reviewer SyrL Q2," you mentioned, "These scale divisions follow the COCO official metrics (Small: area < 32^2, Medium: 32^2 < area < 64^2, Large: area > 64^2)." However, according to the official COCO standards, the criteria for AP Across Scales are:
>
> * AP$^{small}$: AP for small objects: area < 32$^2$;
>
> * AP$^{medium}$: AP for medium objects: 32$^2$ < area < 96$^2$;
>
> * AP$^{large}$: AP for large objects: area > 96$^2$.
>
> It seems that your evaluation criteria are inconsistent with the COCO official standards. What are the specific experimental settings you used? Did you propose a new evaluation standard?  I have doubts about the accuracy of the current experimental results and the fairness of the experimental comparisons.
>
> Reference:
>
>  [R1] Lin T Y, Maire M, Belongie S, et al. Microsoft coco: Common objects in context[C]//Computer Vision–ECCV 2014: 13th European Conference, Zurich, Switzerland, September 6-12, 2014, Proceedings, Part V 13. Springer International Publishing, 2014: 740-755.
>
> 2.  Another small suggestion is regarding the obvious error in the reference name: "NeurIPS" should be used instead of "NIPS."
>
> 3.  There is garbled text in lines 366-367 of the main paper.

---

> > ### Author Response · Authors · 2024-11-24
> > **Response to Reviewer p9Ls: new Q1, Q2, Q3**
> >
> > Thank you for your detailed and active engagement with our paper. We appreciate your effort to ensure the accuracy of our evaluation criteria.
> >
> > >Q1: It seems that your evaluation criteria are inconsistent with the COCO official standards. What are the specific experimental settings you used? Did you propose a new evaluation standard? I have doubts about the accuracy of the current experimental results and the fairness of the experimental comparisons.
> >
> > Thank you for your careful review and for bringing this important point to our attention.
> >
> > 1. You are absolutely correct about the COCO evaluation metrics. We would like to apologize that there was a typo in our response to "Reviewer SyrL Q2" on OpenReview. As correctly documented in our paper's Appendix C.2, all our experiments adhere to these standard COCO evaluation criteria. We have now corrected the typo in our OpenReview response.
> >
> > 2. As noted in our paper (**Appendix C.3. Experiments Settings**), our training and testing protocols rigorously follow the MMDetection [a] framework. Specifically, we use the `coco_metric.py` script within mmdetection for evaluation, where the `COCOeval` function is defined as `COCOeval = _COCOeval` and imported from `pycocotools.cocoeval` as follows:
> >     ```
> >     from pycocotools.cocoeval import COCOeval as _COCOeval
> >     ```
> >     The `pycocotools` package defines the scale divisions exactly as per the COCO standards:
> >     ```
> >     self.areaRng = [[0 ** 2, 1e5 ** 2], [0 ** 2, 32 ** 2], [32 ** 2, 96 ** 2], [96 ** 2, 1e5 ** 2]]
> >     self.areaRngLbl = ['all', 'small', 'medium', 'large']
> >     ```
> >     We strictly follow this code in our evaluations to ensure that our experimental results are accurate and consistent with COCO standards.
> >
> > [a] Chen, et al. "MMDetection: Open mmlab detection toolbox and benchmark." arXiv preprint arXiv:1906.07155 (2019).
> >
> > 3. We will open-source our code soon to provide full transparency and facilitate further verification of our method.
> >
> > > Q2: Another small suggestion is regarding the obvious error in the reference name: "NeurIPS" should be used instead of "NIPS."
> >
> > > Q3: There is garbled text in lines 366-367 of the main paper.
> >
> > Thank you for your careful proof reading and valuable suggestion regarding our paper.
> >
> > 1. We have updated all our rebuttal responses to "NeurIPS" in accordance with your suggestion.
> >
> > 2. For the garbled text in lines 366-367 of our paper, we have corrected this in the manuscript.
> >
> > 3. Additionally, we have also reviewed the main body of our paper to ensure the consistency.
> >
> > Your valuable suggestions greatly contribute to improving the quality of our paper. If you have any further questions, please don't hesitate to ask. Thank you again for your time and expertise.

---

> > > ### Comment · Reviewer_p9Ls · 2024-11-24
> > > **Thanks for the reply**
> > >
> > > I will rise my vote into 6.

---

### Official Review · Reviewer_TLfw · 2024-11-03

**Soundness:** 3
**Presentation:** 3
**Contribution:** 3
**Rating:** 6
**Confidence:** 4

**Summary:**

This paper introduces a training-free dataset pruning method specifically for instance segmentation for the first time. The method leverages object shape information to propose two scores, SI-SCS and CB-SCS, which are then used to select the most challenging samples from the dataset. The method outperforms training-based approaches across different network architectures and datasets.

**Strengths:**

The writing in the paper is clear and easy to understand.
The paper provides thorough experiments across multiple datasets and network architectures.

**Weaknesses:**

1. It is too simple to set the random sample selection strategy as the training-free baseline. Other intuitive selection methods should be compared as baselines to understand better why the proposed method is effective. For example, select images with the most objects or those containing rare categories.

2. The minimum training unit for instance segmentation is the object, not the image. In the proposed dataset pruning task for instance segmentation, is it appropriate to define the pruning ratio based on the number of images? It is suggested that the authors report
both image-level and instance-level pruning ratios for each experiment, and discuss how this affects the interpretation of the results.

3. Does the method show performance improvement over the baseline for all categories? It is suggested that the authors provide a category-wise performance breakdown, highlighting any categories where the proposed method underperforms the baselines, and analyzing potential reasons for these discrepancies.

4. In addition to the random strategy, results at high pruning rates should also include other training-based adaptation methods.

**Questions:**

The main concern is the rationality of the proposed task definition. As shown in Figure 6, the selected images predominantly contain a high number of objects, which implies that the actual object pruning rate could be smaller than the corresponding image pruning rate. The comparison between averaging image scores and summation image scores in the paper also supports this point. From this perspective, the comparison of the proposed method with other baseline methods may be unfair. The authors are suggested to:
1. Provide a detailed analysis of how the image-level pruning translates to instance-level pruning.
2. Discuss the implications of this discrepancy on the results and comparisons.
3. Conduct additional experiments by controlling the total number of object instances rather than images when comparing methods. This would help address the potential bias towards selecting images with many objects and provide a more equitable comparison.

---

> ### Author Response · Authors · 2024-11-22
> **Rebuttal 1: W1**
>
> Thank you for your insightful feedback. Your positive remarks about our work being "clear and easy to understand" and featuring "thorough experiments" are highly motivating.
>
> Additionally, the weaknesses and questions you highlighted have been very instructive. We have carefully considered the issues you raised and offer the responses below.
>
> > W1: It is too simple to set the random sample selection strategy as the training-free baseline. Other intuitive selection methods should be compared as baselines to understand better why the proposed method is effective. For example, select images with the most objects or those containing rare categories.
>
> Thank you for your valuable suggestions.
>
> **We have added several strong training-free baselines** on COCO dataset for reference, including:
> - **K-Means Clustering:** Groups instances based on category features by selecting cluster centroids and their neighboring data points (sorted by Euclidean distance) as representative data points.
> - **Herding [a]:** Select data by dynamically adjusting weights to prioritize constraint satisfaction and efficient estimation.
> - **Instance Count-Based Pruning** **(Count):** Prioritizes images with a higher number of instances within a single image for selection. *Notably, the final step of our proposed TFDP method involves summing the scores of all instances within an image, inherently considering the instance count. Therefore, **instance count-based pruning is essentially a special variant of our method**, where Count assumes all instance scores are equal to 1.*
> - **Category Balance-Based Pruning** **(Category):** Selects images from each category using SI-SCS, ensuring an equal number of instances per category. If a category runs out of instances at the current pruning rate, the selection continues with the remaining categories.
>
> Table B1. More training-free baseline comparison results.
>
> | pruning rate | 20% COCO (mAP/AP50/AP75) | 30% COCO (mAP/AP50/AP75) | 40% COCO (mAP/AP50/AP75) | 50% COCO (mAP/AP50/AP75) |
> | ------------ | ------------------------ | ------------------------ | ------------------------ | ------------------------ |
> | K-means      | 32.3/52.7/35.0           | 31.9/52.2/33.9           | 31.3/51.1/33.4           | 30.5/50.2/32.0           |
> | Herding      | 32.0/51.8/34.6           | 31.1/49.9/33.4           | 30.0/48.4/32.1           | 29.8/48.3/31.4           |
> | Count        | 33.1/54.3/35.5           | 32.8/53.4/34.3           | 32.0/52.9/34.1           | 31.0/51.4/32.8           |
> | Category     | 31.3/51.5/33.8           | 30.9/49.6/33.2           | 30.1/48.4/32.2           | 29.6/47.8/31.0           |
> | Ours         | **34.4/55.5/36.7**       | **33.6/54.8/35.4**       | **33.1/54.2/35.1**       | **32.5/53.4/34.3**       |
>
> The results indicate that our method **maintains a clear advantage** compared to all these new strong baselines. Additionally, the results show that Count demonstrates good performance. However, further analysis reveals that Count performs poorly under high pruning rates, falling below random selection. In contrast, our method remains robust in such scenarios. (Please refer to the answer to Weakness 4 (W4): Results of other training-based baselines at high pruning rates.)
>
> Moreover, we would like to clarify that in dataset pruning tasks for image classification, **Random has been demonstrated to be a stable and robust baseline** [b]**.** This is also reflected in our experiments: many baseline methods perform worse than Random under certain settings. Random's strength lies in its ability to evenly cover the entire data distribution, ensuring that the selected data effectively represents the full dataset to some extent.
>
> [a] Welling, Max. Herding dynamical weights to learn. _ICML_ _(2009)._
>
> [b] Zheng, et al. Coverage-centric coreset selection for high pruning rates. ICLR (2023).

---

> ### Author Response · Authors · 2024-11-22
> **Rebuttal 2: W2**
>
> >W2:  The minimum training unit for instance segmentation is the object, not the image. In the proposed dataset pruning task for instance segmentation, is it appropriate to define the pruning ratio based on the number of images? It is suggested that the authors report both image-level and instance-level pruning ratios for each experiment, and discuss how this affects the interpretation of the results.
>
> Thank you for your valuable insights. As mentioned in Sec. 3.1 of our paper: "We reduce the number of images (image-level) rather than instance annotations (instance-level) since the large volume of image data primarily impacts training time and storage requirements."
>
> To further analyze and explain the issues at both the instance-level and image-level, we provide the following detailed analysis:
>
> 1.**The impact of the number of instances on training time is small.** Our analysis focused on three main training stages: network forward, loss calculation, and backpropagation. We compared training times using two sets of annotations within the same batch (batch size = 8): one set with complete annotations totaling 78 instances and another set pruned to just 8 instances (one per image).
>
> Table B2. Training time breakdown experiment (Averaged over three trials).
>
> | Time (seconds)                  | Network forward | Loss Calculation | Backpropagation | Total        |
> | ------------------------------- | --------------- | ---------------- | --------------- | ------------ |
> | Full annotation (78 instances)  | 1.18 seconds    | 0.09 seconds     | 1.45 seconds    | 2.72 seconds |
> | Pruned annotation (8 instances) | 1.18 seconds    | 0.04 seconds     | 1.43 seconds    | 2.65 seconds |
>
> The results in Table B2 show that differences in training time are negligible despite substantial differences in instance count. Notably, the slight increase in loss calculation time for more instances is minor compared to the extensive computations required for the network forward and backpropagation stages. These stages, which involve processing the images through the network and updating all network parameters, are largely independent of the number of instances.
>
> 2.**In segmentation datasets, the storage** **of** **images is significantly greater than that of labels.** As shown in the table below, since label files are generally stored in a text format (.json), their storage is significantly smaller than the images.
>
> Table B3. Comparison of storage.
>
> |               | VOC     | Cityscapes | COCO    |
> | ------------- | ------- | ---------- | ------- |
> | Image storage | 1.9 GB  | 11 GB      | 20 GB   |
> | Label storage | 0.07 GB | 0.24 GB    | 0.99 GB |

---

> ### Author Response · Authors · 2024-11-22
> **Rebuttal 3: W2-continue**
>
> 3.**Comparison of training time for different methods.** We compared random pruning (Random), pruning by instance count (Count), and our method. As shown in the table below, we draw the following significant conclusions. First, different methods will lead to different instance pruning rates even if the image pruning rate is the same. Second, **the number of images dominates the training time**, while instance pruning rates have little influence on training time.
>
> Table B4. Comparison of pruning rate, training time and performance.
>
> | Method           | Image Pruning Rate | Corresponding Instance Pruning Rate | Training Time | Time Accelateion Ratio | Performance (%) (mAP/AP50/AP75) |
> | ---------------- | ------------------ | ----------------------------------- | ------------- | ---------------------- | ------------------------------- |
> | Full (0% images) | 0%                 | 0%                                  | 12hrs 45mins  | 0%                     | 34.2/55.2/36.5                  |
> | Random           | 20%                | 20.15%                              | 10hrs 18mins  | 19.21%                 | 33.6/54.5/35.6                  |
> |                  | 30%                | 29.99%                              | 9hrs 40mins   | 24.18%                 | 32.1/52.8/34.1                  |
> |                  | 40%                | 39.92%                              | 8hrs 16mins   | 35.16%                 | 31.1/51.1/33.2                  |
> |                  | 50%                | 49.88%                              | 7hrs 21mins   | 42.35%                 | 30.8/51.0/32.7                  |
> | Count            | 20%                | 3.89%                               | 10hrs 42mins  | 16.08%                 | 33.9/55.0/36.1                  |
> |                  | 30%                | 6.64%                               | 9hrs 53mins   | 22.48%                 | 32.9/54.0/34.9                  |
> |                  | 40%                | 10.78%                              | 8hrs 27mins   | 33.73%                 | 32.0/52.9/34.1                  |
> |                  | 50%                | 16.01%                              | 7hrs 38mins   | 40.13%                 | 31.0/51.4/32.8                  |
> | **Ours**         | 20%                | 6.57%                               | 10hrs 29mins  | 17.78%                 | **34.4/55.5/36.7**              |
> |                  | 30%                | 11.29%                              | 9hrs 38mins   | 24.45%                 | **33.2/54.4/35.3**              |
> |                  | 40%                | 17.26%                              | 8hrs 20mins   | 34.64%                 | **33.1/54.2/35.1**              |
> |                  | 50%                | 24.27%                              | 7hrs 22mins   | 42.22%                 | **32.5/53.4/34.3**              |

---

> ### Author Response · Authors · 2024-11-22
> **Rebuttal 4: W2-continue**
>
> 4.Comparison of Image and Instance Pruning Rates: We evaluated various metrics under different image-level (Table B5) and instance-level (Table B6) pruning settings, including their respective pruning rates, training times, and performance outcomes. The Tables show that 1) **Training Time Influences**: The number of images significantly impacts training time. 2) **Performance Comparison**: Our method consistently surpasses the Random baseline in both image-level and instance-level settings. 3) **Pruning Efficiency**: Image-level pruning is more efficient than instance-level. The speed improvement from instance-level pruning is quite limited.
>
> Table B5. Image-Level pruning results. Under the same image pruning rate, differences are small (up to a 24-minutes variance).
>
> | method | image pruning rate | corresponding instance pruning rate | Training Time | Performance (%) (mAP/AP50/AP75) |
> | ------ | ------------------ | ----------------------------------- | ------------- | ------------------------------- |
> | Random | 20% images         | 20.15% instances                    | 10hrs 18mins  | 33.6/54.5/35.6                  |
> |        | 30% images         | 30.90% instances                    | 9hrs 29mins   | 32.1/52.8/34.1                  |
> |        | 40% images         | 40.60% instances                    | 8hrs 26mins   | 31.1/51.1/33.2                  |
> |        | 50% images         | 49.88% instances                    | 7hrs 21mins   | 30.8/51.0/32.7                  |
> | Ours   | 20% images         | 6.57% instances                     | 10hrs 42mins  | **34.4/55.5/36.7**              |
> |        | 30% images         | 12.33% instances                    | 9hrs 34mins   | **33.6/54.8/35.5**              |
> |        | 40% images         | 18.22% instances                    | 8hrs 21mins   | **33.1/54.2/35.1**              |
> |        | 50% images         | 24.27% instances                    | 7hrs 22mins   | **32.5/53.4/34.3**              |
>
> Table B6. Instance-Level pruning results. Under the same instance pruning rate, differences can be substantial (up to 3 hours and 9 minutes).
>
> | method | instance pruning rate | corresponding image pruning rate | Training Time | Performance (%) (mAP/AP50/AP75) |
> | ------ | --------------------- | -------------------------------- | ------------- | ------------------------------- |
> | Random | 20% instances         | 4.05% images                     | 11hrs 10mins  | 33.1/53.8/35.3                  |
> |        | 30% instances         | 6.50% images                     | 10hrs 44mins  | 32.3/52.6/34.3                  |
> |        | 40% instances         | 9.50% images                     | 10hrs 04mins  | 31.7/51.1/33.5                  |
> |        | 50% instances         | 13.59% images                    | 9hrs 24mins   | 30.8/50.2/32.5                  |
> | Ours   | 20% instances         | 0.15% images                     | 12hrs 38mins  | **34.6/55.7/36.8**              |
> |        | 30% instances         | 0.30% images                     | 12hrs 25mins  | **34.4/55.3/36.9**              |
> |        | 40% instances         | 0.52% images                     | 12hrs 21mins  | **33.5/54.3/35.8**              |
> |        | 50% instances         | 0.76% images                     | 12hrs 23mins  | **32.4/51.7/34.4**              |

---

> ### Author Response · Authors · 2024-11-22
> **Rebuttal 5: W3, W4, Q1**
>
> >W3: Does the method show performance improvement over the baseline for all categories? It is suggested that the authors provide a category-wise performance breakdown, highlighting any categories where the proposed method underperforms the baselines, and analyzing potential reasons for these discrepancies.
>
> Thank you for your valuable suggestions. We have further demonstrated and analyzed the performance across all categories on Cityscapes at a pruning rate of 50%. As shown in the table below, our method outperforms the baseline methods in all categories, further highlighting the superiority of our approach. *The results for VOC (20 classes) and COCO (80 classes) will be updated in the appendix* due to space limitations.
>
> Table B7. Category-wise performance results.
>
> | Cityscapes (p=50%) | person   | rider    | car      | truck    | bus      | train    | motorcycle | bicycle  | Overall  |
> | ------------------ | -------- | -------- | -------- | -------- | -------- | -------- | ---------- | -------- | -------- |
> | Random             | 33.3     | 27.0     | 51.9     | 33.7     | 52.2     | 34.9     | 19.8       | 20.3     | 33.8     |
> | EL2N               | 33.9     | 26.3     | 51.9     | 31.6     | 51.0     | 23.1     | 20.9       | 19.7     | 32.1     |
> | AUM                | 35.1     | 27.0     | 52.1     | 33.2     | 55.3     | 33.2     | 18.6       | 20.5     | 33.9     |
> | CCS                | 35.2     | 27.6     | 52.0     | 34.0     | 52.5     | 35.6     | 20.8       | 21.3     | 34.0     |
> | Ours               | **36.3** | **29.5** | **52.2** | **36.4** | **55.4** | **39.3** | **20.3**   | **23.2** | **36.6** |
>
> > W4: In addition to the random strategy, results at high pruning rates should also include other training-based adaptation methods.
>
> Thank you for your insightful suggestions.
>
> We have added comparisons of other methods at high pruning rates. The table below shows that our method still **significantly outperforms the baselines**. Notably, as the pruning rate increases, the performance of many baseline methods declines significantly, consistent with conclusions drawn from pruning methods in classification tasks [a]. It is worth noting that we also compared the new baseline (Herding, Count, and Category) discussed in Weakness 1 (W1).
>
> Table B8. High pruning rate results.
>
> |pruning rate|60% (mAP/AP50/AP75)|70% (mAP/AP50/AP75)|80% (mAP/AP50/AP75)|90% (mAP/AP50/AP75)|
> |---|---|---|---|---|
> |Random|30.4/49.8/32.3|29.0/48.4/30.6|26.6/45.6/27.5|21.6/38.9/22.5|
> |EL2N|28.7/47.4/31.2|27.1/46.2/28.3|22.1/40.5/23.7|17.3/34.6/18.3|
> |AUM|26.3/45.5/27.6|24.9/44.6/25.7|22.0/40.3/23.5|16.8/33.3/17.8|
> |CCS|30.2/51.6/31.4|28.2/48.1/28.8|26.4/46.7/26.8|21.4/38.5/22.3|
> |Herding|28.4/47.1/29.8|26.6/45.7/27.8|22.9/41.5/23.2|18.4/34.4/20.2|
> |Count|29.8/50.2/31.2|27.9/47.6/28.4|24.3/43.4/24.5|17.9/33.9/17.0|
> |Category|28.4/47.0/30.1|27.7/46.9/29.2|26.7/45.9/24.9|19.9/37.1/20.1|
> |Ours|**31.3/52.1/32.8**|**29.6/50.4/30.7**|**27.0/46.8/27.6**|**21.8/39.8/22.5**|
>
> [a] Zheng, et al. Coverage-centric coreset selection for high pruning rates. ICLR (2023).
>
> >Q1:  The main concern is the rationality of the proposed task definition. As shown in Figure 6, the selected images predominantly contain a high number of objects, which implies that the actual object pruning rate could be smaller than the corresponding image pruning rate. The comparison between averaging image scores and summation image scores in the paper also supports this point. From this perspective, the comparison of the proposed method with other baseline methods may be unfair.
>
> Thank you for your attention to this issue, which has been instrumental in refining our paper.
>
> We have provided detailed results and analysis in response to your **Weakness 2** **(W2)**.
>
> Overall, image-level pruning is evidently more reasonable than instance-level pruning. This is primarily because the number of images dictates the training time. 1) Under the **image-level pruning setting**, the training resources and time consumed by various baselines are similar, making it a **fair** comparison. 2) Under the **instance-level pruning**, the corresponding image-level pruning rate is uncertain, which leads to significant variations in training time. This results in **unfair** comparisons as training resources (time) become a new, uncontrollable variable.

---

> ### Comment · Reviewer_TLfw · 2024-11-24
> **Response**
>
> Thanks for your response. After reading the response and other reviewers' reviews, I'd like to raise my score to 6.

---

### Official Review · Reviewer_SyrL · 2024-11-03

**Soundness:** 3
**Presentation:** 3
**Contribution:** 2
**Rating:** 6
**Confidence:** 3

**Summary:**

The paper presents a novel Training-Free Dataset Pruning (TFDP) method designed for instance segmentation tasks. Unlike existing dataset pruning techniques that focus on image classification and require time-consuming model training, TFDP leverages shape and class information from image annotations to prune datasets without any model training. The authors introduce a Shape Complexity Score (SCS), which is further refined into Scale-Invariant (SI-SCS) and Class-Balanced (CB-SCS) versions to address the challenges of instance area variations and class imbalances. This approach is intuitive and appears effective, with promising results.

**Strengths:**

- This approach is intuitive and appears effective
- This is the first work to address dataset pruning for instance segmentation.
- By eliminating the need for model training, TFDP aligns well with the goal of dataset pruning, which is to reduce resource consumption. And significantly accelerates the pruning process.
- The paper is well-structured, with clear and concise explanations supported by intuitive figures.

**Weaknesses:**

- Given that instance segmentation tasks are often data-scarce due to the high cost of annotations, the immediate practical utility of dataset pruning in this domain may be limited.
- As prior work on instance segmentation pruning is lacking, the authors’ adaptation of classification-oriented methods may make baseline comparisons less compelling.

**Questions:**

Some thoughts and explorations:
1. Impact of Annotation Quality: The masks in the tested datasets are manually annotated, implying potential inaccuracies, especially for smaller objects. Since SI-SCS may assign lower scores to smaller objects due to annotation errors, it could lead to a bias towards pruning these objects. Given that the mean average precision (mAP) metric is averaged over instances, pruning smaller objects could disproportionately affect the overall mAP score.
2. Balancing Scale Diversity: While SI-SCS aims to make the score scale-invariant and CB-SCS balances class contributions, it would be interesting to explore whether incorporating additional mechanisms to balance the pruning of objects across different scales could further improve performance, particularly in scenarios where small objects are critical for overall accuracy.

---

> ### Author Response · Authors · 2024-11-22
> **Rebuttal 1 : W1**
>
> Thank you for your constructive comments. We are greatly encouraged by your appreciation of our work as "novel", "intuitive" and "effective".
>
> You also raised some points of weakness and questions, which we found very enlightening. We thoroughly analyzed the issues you mentioned and provided the following responses.
>
> > W1:  Given that instance segmentation tasks are often data-scarce due to the high cost of annotations, the immediate practical utility of dataset pruning in this domain may be limited.
>
> 1.**Our proposed method not only reduces training time but also improves model performance.** According to the experimental results below, our approach exceeds the performance of full data (mAP 34.2%) with only 80% (mAP 34.4%) of the data on the COCO dataset. Furthermore, on the Cityscapes dataset, our method surpasses the full dataset performance (mAP 36.4%) with merely 20% data (mAP 36.9%).
>
> Table A1: Improved performance at a low pruning rate.
>
> | Method   | pruning rate      | COCO (mAP/AP50) | Cityscapes (mAP/AP50) |
> | -------- | ----------------- | --------------- | --------------------- |
> | Baseline | 0% (Full dataset) | 34.2/55.2       | 36.4/61.8             |
> | Ours     | 20%               | **34.4/55.5**   | **36.9/62.8**         |
>
> 2.The improvement may come from the removal of noisy samples or samples with annotation errors, leading to better dataset quality. Therefore, our method has the potential to examine the annotation quality.
>
> 3.Dataset pruning research in classification tasks has evolved from small to large-scale datasets. Early pruning works [a,b,c] focused on CIFAR-10, while recent studies [d] have successfully scaled their methods to ImageNet-1k. Meanwhile, instance segmentation datasets have now reached an unprecedented scale, with SA-1B significantly surpassing ImageNet-1k in size.
>
> As a pioneering work in this domain, while our approach currently demonstrates strong performance on mainstream datasets like COCO, its design principles make it particularly promising for future applications on ultra-large-scale datasets such as SA-1B. We have compared these datasets with those used for segmentation tasks regarding dataset size, as illustrated in the table below.
>
>
> Table A2: Summary of classification and segmentation dataset.
>
> |               | CIFAR-10       | CIFAR-100      | Tiny-IN        | ImageNet-1k    | VOC          | Cityscapes   | COCO         | SA-1B        |
> | ------------- | -------------- | -------------- | -------------- | -------------- | ------------ | ------------ | ------------ | ------------ |
> | Type          | Classification | Classification | Classification | Classification | Segmentation | Segmentation | Segmentation | Segmentation |
> | Image size    | 60 K           | 60 K           | 120 K          | 1.28 M         | 1.5 K        | 5 K          | 330 K        | 11M          |
> | Image Storage | 0.16 GB        | 0.16 GB        | 0.48 GB        | 128 GB         | 1.9 GB       | 11 GB        | 20 GB        | 10,000 GB    |
> | Label size    | -              | -              | -              | -              | 17 K         | 60 K         | 1.5 M masks  | 1.1B masks   |
> | Label Storage | -              | -              | -              | -              | 0.06 GB      | 0.23 GB      | 0.99 GB      | ~726GB       |
>
> [a] Toneva, et al. An empirical study of example forgetting during deep neural network learning. ICLR (2019).
>
> [b] Paul, et al. Deep learning on a data diet: Finding important examples early in training. NeurIPS (2021).
>
> [c] Pleiss, et al. Identifying mislabeled data using the area under the margin ranking. NeurIPS (2020).
>
> [d] Zheng, et al. Coverage-centric coreset selection for high pruning rates. ICLR (2023).

---

> ### Author Response · Authors · 2024-11-22
> **Rebuttal 2: W2**
>
> > W2: As prior work on instance segmentation pruning is lacking, the authors’ adaptation of classification-oriented methods may make baseline comparisons less compelling.
>
> Thank you for your valuable suggestions.  Apart from classification-oriented pruning methods, we include additional strong baselines for comprehensive comparison.
>
> - **K-means Clustering:** Groups instances based on category features by selecting cluster centroids and their neighboring data points (sorted by Euclidean distance) as representative data points.
> - **Herding** **[e]:** Select data by dynamically adjusting weights to prioritize constraint satisfaction and efficient estimation. It has been widely used by papers on data efficiency [f,g,h,i].
> - **Instance Count-Based Pruning** **(Count):** Prioritizes images with a higher number of instances within a single image for selection.
> - **Category Balance-Based Pruning** **(Category):** Selects images from each category using SI-SCS, ensuring an equal number of instances per category. If a category runs out of instances at the current pruning rate, the selection continues with the remaining categories.
>
> Table A3: More baselines comparison results.
>
> | Pruning Rate | 20% COCO (mAP/AP50/AP75) | 30% COCO (mAP/AP50/AP75) | 40% COCO (mAP/AP50/AP75) | 50% COCO (mAP/AP50/AP75) |
> | ------------ | ------------------------ | ------------------------ | ------------------------ | ------------------------ |
> | K-means      | 32.3/52.7/35.0           | 31.9/52.2/33.9           | 31.3/51.1/33.4           | 30.5/50.2/32.0           |
> | Herding      | 32.0/51.8/34.6           | 31.1/49.9/33.4           | 30.0/48.4/32.1           | 29.8/48.3/31.4           |
> | Count        | 33.1/54.3/35.5           | 32.8/53.4/34.3           | 32.0/52.9/34.1           | 31.0/51.4/32.8           |
> | Category     | 31.3/51.5/33.8           | 30.9/49.6/33.2           | 30.1/48.4/32.2           | 29.6/47.8/31.0           |
> | Ours         | **34.4/55.5/36.7**       | **33.6/54.8/35.4**       | **33.1/54.2/35.1**       | **32.5/53.4/34.3**       |
>
> The results indicate that our method **maintains a clear advantage** compared to all these new strong baselines.
>
> [e] Welling, Max. Herding dynamical weights to learn. ICML (2009).
>
> [f] Kim, et al. Dataset condensation via efficient synthetic-data parameterization. ICML (2022)
>
> [g] Huang, et al. Coresets for clustering with fairness constraints. NeurIPS (2019)
>
> [h] Braverman, et al. Coresets for ordered weighted clustering. ICML (2019)
>
> [i]  Xia , et al. Moderate Coreset: A Universal Method of Data Selection for Real-world Data-efficient Deep Learning. ICLR (2023)

---

> ### Author Response · Authors · 2024-11-22
> **Rebuttal 3: Q1**
>
> > Q1: Impact of Annotation Quality: The masks in the tested datasets are manually annotated, implying potential inaccuracies, especially for smaller objects. Since SI-SCS may assign lower scores to smaller objects due to annotation errors, it could lead to a bias towards pruning these objects. Given that the mean average precision (mAP) metric is averaged over instances, pruning smaller objects could disproportionately affect the overall mAP score.
>
> Thank you for your insightful observation. Our SI-SCS module's scoring mechanism, based on Equation 12, does tend to assign lower scores to smaller objects based on their area than SCS. However, this behavior is intentionally designed - it selectively retains only the most informative small objects for training. This selective approach actually helps mitigate annotation quality issues, as small objects with high scores typically indicate more precise and reliable annotations. In effect, this scoring mechanism naturally filters out potentially noisy or imprecise small object annotations.
>
> To further address your concerns, we use the tables below from our manuscript to present all COCO official evaluation metrics [i], including mAP, AP50, AP75, as well as scale-related metrics such as AP_S, AP_M, and AP_L.
>
> Table A4. (Manuscript Page 8, Table 1): The mask AP (%) results for different IoU threshold.
>
> | pruning rate | 20% COCO (mAP/AP50/AP75) | 30% COCO (mAP/AP50/AP75) | 40% COCO (mAP/AP50/AP75) | 50% COCO (mAP/AP50/AP75) |
> | ------------ | ------------------------ | ------------------------ | ------------------------ | ------------------------ |
> | Random       | 33.6/54.5/35.6           | 32.1/52.8/34.1           | 31.1/51.1/33.2           | 30.8/51.0/32.7           |
> | Forgetting   | 33.1/54.2/35.2           | 32.3/53.4/34.3           | 31.4/52.2/33.4           | 30.4/51.2/32.1           |
> | Entropy      | 33.2/54.4/35.5           | 32.3/53.5/34.5           | 31.4/52.5/33.2           | 30.9/51.7/32.6           |
> | EL2N         | 33.4/54.5/35.6           | 32.1/52.9/34.2           | 31.2/51.7/33.2           | 30.5/51.2/32.0           |
> | AUM          | 33.5/54.6/35.5           | 32.4/53.3/34.7           | 31.5/52.4/33.4           | 31.0/51.7/32.8           |
> | CCS          | 33.4/54.1/35.6           | 32.4/53.3/34.4           | 31.7/52.6/33.6           | 31.5/52.3/33.2           |
> | Ours         | **34.4/55.5/36.7**       | **33.6/54.8/35.4**       | **33.1/54.2/35.1**       | **32.5/53.4/34.3**       |
>
> Table A5. (Manuscipt Page 20, Table 8): The mask AP (%) results for different object areas (Small, Medium, Large) on COCO dataset.
>
> | pruning rate | 20% COCO (AP_S/AP_M/AP_L) | 30% COCO (AP_S/AP_M/AP_L) | 40% COCO (AP_S/AP_M/AP_L) | 50% COCO (AP_S/AP_M/AP_L) |
> | ------------ | ------------------------- | ------------------------- | ------------------------- | ------------------------- |
> | Random       | 15.9/36.0/48.8            | 14.1/34.6/48.1            | 13.1/33.2/46.2            | 13.2/33.2/45.6            |
> | Forgetting   | 15.6/35.7/49.1            | 15.0/35.1/46.5            | 14.6/34.2/44.9            | 14.5/33.7/43.6            |
> | Entropy      | 15.5/35.9/48.9            | 14.9/35.0/46.7            | 14.3/34.1/45.1            | 14.4/33.9/43.8            |
> | EL2N         | 15.6/36.3/47.9            | 14.5/35.2/35.2            | 14.4/34.1/44.6            | 14.4/33.6/43.2            |
> | AUM          | 15.8/36.3/48.5            | 14.7/35.4/46.6            | 14.6/34.3/45.3            | 14.4/34.1/44.0            |
> | CCS          | 15.8/36.0/48.4            | 15.2/34.9/46.7            | 14.8/34.6/45.0            | 14.7/34.1/45.2            |
> | Ours         | **16.2/37.1/49.3**        | **15.5/36.3/48.5**        | **15.5/36.0/46.9**        | **15.1/35.3/46.2**        |
>
> The results in the table show that our method consistently outperforms the baselines across all metrics. More importantly, the performance improvement on small objects (AP_S) validates the effectiveness of SI-SCS.

---

> ### Author Response · Authors · 2024-11-22
> **Rebuttal 4: Q2**
>
> > Q2: Balancing Scale Diversity: While SI-SCS aims to make the score scale-invariant and CB-SCS balances class contributions, it would be interesting to explore whether incorporating additional mechanisms to balance the pruning of objects across different scales could further improve performance, particularly in scenarios where small objects are critical for overall accuracy.
>
> Your question is very insightful. We have explored it by comparing the following **scale-balancing selection** methods:
>
> 1. **Specific scale distribution**: Select scales that best preserve the original distribution of the training set.
> 2. **Uniform scale distribution**: Choose the same number of images from each scale division (Small, Medium, Large).
>
> These scale divisions follow the COCO official metrics [j] (Small: area < 32^2, Medium: 32^2 < area < 96^2, Large: area > 96^2). Notably, both the **COCO (S: 40.35%, M: 35.31%, L: 24.34%)** and **Cityscapes (S: 41.35%, M: 40.03%, L: 18.62%)** datasets have a relatively **high proportion of small instance,** which are critical for overall accuracy in these scenarios.
>
> Table A6. Experiments on different scale balancing strategies.
>
> |          | 50% Cityscapes (mAP/AP50/AP75) | 50% COCO (mAP/AP50/AP75) |
> | -------- | ------------------------------ | ------------------------ |
> | Specific | 34.7/60.6/33.8                 | 32.0/52.5/33.1           |
> | Uniform  | 33.4/58.6/33.6                 | 31.3/51.6/32.4           |
> | Ours     | **36.6/62.8/35.4**             | **32.5/53.4/34.3**       |
>
> The results indicate that our proposed method for eliminating scale bias **performs best**. This is primarily because the scale bias in SCS arises from a combination of the dataset's scale diversity and the inherent small area bias of SCS. Our proposed Scale Invariance (SI) design directly normalizes the scales in SCS, resulting in SI-SCS. In contrast, the two scale-balancing methods (Specific and Balanced) mitigate scale imbalance by stratifying and filtering SCS based on scale, but they fail to address the issue entirely. Moreover, we do **not need** additional scale distribution information to make our selections.
>
> Additionally, your suggestion is highly valuable. For example, in autonomous driving, small objects like pedestrians and traffic signs are critical for accuracy. This direction holds great potential for improving performance in such scale-sensitive applications and represents a compelling area for our future research.
>
> [j] Lin, et al. Microsoft coco: Common objects in context. ECCV (2014).

---

> > ### Comment · Reviewer_SyrL · 2024-11-25
> > **Will raise the rating to 6**
> >
> > Thank you for your detailed response, which has helped clarify my doubts. I will raise the rating to 6 for the following reasons:
> >
> > - Given that this is a new task and the authors have provided a sufficiently rich set of baselines, it is a good start.
> > - Although I have doubts about whether the contribution of the authors' proposed Shape Complexity Score meets the quality standards of ICLR, this  technique is still worth sharing with a broader audience.

---

### Meta-Review · Area_Chair_UuZy · 2024-12-23

**Metareview:**

The paper introduces a training-free dataset pruning method specifically tailored for instance segmentation tasks, which uses shape and class information from annotations to optimize pruning without requiring model training. This is achieved through Shape Complexity Score refinements to address area variations and class imbalances. The method demonstrates efficiency and generalization across architectures and datasets while significantly accelerating the pruning process.

Reviewers appreciated the paper's novelty, clarity, and the comprehensive experimental results. Strengths included its pioneering approach to a new task and its effectiveness in improving performance metrics with reduced data. However, concerns were raised about the lack of detailed explanations in some theoretical derivations, limitations in addressing instance-level pruning in certain comparisons, and the insufficient exploration of certain failure cases and biases. The authors addressed most issues through extensive rebuttals and revisions, improving clarity and adding further experiments.

Considering the novelty of the approach, its performance gains, and the authors' responsiveness to critiques, the AC recommends acceptance as a poster presentation.

**Additional Comments On Reviewer Discussion:**

Please refer to the above metareview.

---

### Decision · Program_Chairs · 2025-01-22

Accept (Poster)